# Cloning and high-level expression of monomeric human superoxide dismutase 1 (SOD1) and its interaction with pyrimidine analogs

Marcia LeVatte[1], Matthias Lipfert[1], Dipankar Roy[2], Andriy Kovalenko[2,3], David Scott Wishart[1,4]*

1 Department of Biological Sciences, University of Alberta, Edmonton, AB, Canada, 2 Department of Mechanical Engineering, University of Alberta, Edmonton, AB, Canada, 3 Nanotechnology Research Centre, Edmonton, AB, Canada, 4 Department of Computing Science, University of Alberta, Edmonton, AB, Canada

* dwishart@ualberta.ca

**Data Availability Statement:** All relevant data are within the paper and its Supporting Information files.

**Funding:** This work was supported by funding from the Alberta Innovates Alberta Prion Research

## Abstract

Superoxide dismutase 1 (SOD1) is known to be involved in the pathogenesis of Amyotrophic Lateral Sclerosis (ALS) and is therefore considered to be an important ALS drug target. Identifying potential drug leads that bind to SOD1 and characterizing their interactions by nuclear magnetic resonance (NMR) spectroscopy is complicated by the fact that SOD1 is a homodimer. Creating a monomeric version of SOD1 could alleviate these issues. A specially designed monomeric form of human superoxide dismutase (T2M4SOD1) was cloned into *E. coli* and its expression significantly enhanced using a number of novel DNA sequence, leader peptide and growth condition optimizations. Uniformly $^{15}$N-labeled T2M4SOD1 was prepared from minimal media using $^{15}$NH$_4$Cl as the $^{15}$N source. The T2M4SOD1 monomer (both $^{15}$N labeled and unlabeled) was correctly folded as confirmed by $^1$H-NMR spectroscopy and active as confirmed by an in-gel enzymatic assay. To demonstrate the utility of this new SOD1 expression system for NMR-based drug screening, eight pyrimidine compounds were tested for binding to T2M4SOD1 by monitoring changes in their $^1$H NMR and/or $^{19}$F-NMR spectra. Weak binding to 5-fluorouridine (FUrd) was observed via line broadening, but very minimal spectral changes were seen with uridine, 5-bromouridine or trifluridine. On the other hand, $^1$H-NMR spectra of T2M4SOD1 with uracil or three halogenated derivatives of uracil changed dramatically suggesting that the pyrimidine moiety is the crucial binding component of FUrd. Interestingly, no change in tryptophan 32 (Trp32), the putative receptor for FUrd, was detected in the $^{15}$N-NMR spectra of $^{15}$N-T2M4SOD1 when mixed with these uracil analogs. Molecular docking and molecular dynamic (MD) studies indicate that interaction with Trp32 of SOD1 is predicted to be weak and that there was hydrogen bonding with the nearby aspartate (Asp96), potentiating the Trp32-uracil interaction. These studies demonstrate that monomeric T2M4SOD1 can be readily used to explore small molecule interactions via NMR.

Institute (Research team program ABIBS APRIRTP 201300023 and explorations program ABIBS APRIEP 201600034; https://albertainnovates.ca/programs/alberta-prion-research-institute/). DR and AK acknowledge generous computing time provided by WestGrid (www.westgrid.ca) and Compute Canada/Calcul Canada (www.computecanada.ca).The funders had no role in study design, data collection and analysis, decision to publish, or preparation of the manuscript.

**Competing interests:** The authors have declared that no competing interests exist.

## Introduction

Amyotrophic Lateral Sclerosis (ALS) is a neurodegenerative disease that affects the motor neurons in the brain and spinal cord. The loss of these motor neurons leads to an inability to move, walk, swallow, speak and, eventually, breathe. ALS is incurable and uniformly fatal with the life expectancy of ALS patients often being less than 5 years after diagnosis. Ten percent of ALS cases are inherited (familial ALS, fALS) while the majority (90%) of cases occur sporadically (sALS), with no known cause. Proteins linked to fALS and sALS include the transactive response (TAR) DNA-binding protein 43 (TDP-43), an RNA binding protein called fused in sarcoma/translocated in liposarcoma and copper zinc superoxide dismutase (Cu/Zn SOD1). SOD1 is a small (32 kDa), abundant, homodimeric antioxidant protein found in the mitochondria and cytoplasm. SOD1 mutations are found in a high proportion of fALS cases (20%), with > 180 different dominant mutations mapped to the *sod1* gene [1,2]. There is evidence that the folding or misfolding of Cu/Zn SOD1 [3] and TDP-43 [4] may contribute to sALS, as aggregates of SOD1 have been found in post-mortem neural tissues of sALS individuals. Because patients with fALS and sALS present with similar symptoms, it is thought that common pathological mechanisms may be involved. Therefore, what is learned from studying fALS may aid in the understanding of sALS disease process.

A large body of evidence and an improved understanding of ALS pathogenesis has been gained from studying transgenic rodent models expressing human normal and fALS-SOD1 variants [5]. Many of these studies point to a gain of function of SOD1 that may arise from protein misfolding. Cozzolino et al. [6], demonstrated that a particular fALS mutant of SOD1 (glycine 93 to alanine, G93A) could transfer its aggregation propensity to wild-type SOD1 *in vitro*. In another study, transgenic animals expressing human wild-type human SOD1 (wtSOD1) or SOD1-deficient mice were shown to be resistant to developing ALS. However, inserting genes for a mutant SOD1 fALS gene accelerated the disease process or converted an unaffected phenotype to an ALS clinical and pathological phenotype [7]. These studies suggest that the toxic phenotype can be propagated *in vitro* and *in vivo*. Therefore, preventing wtSOD1 from misfolding and aggregating may prevent the progression of ALS.

SOD1 is a metalloenzyme, consisting of two 153 amino acid subunits. Each subunit is stabilized by an intra disulfide bond between cysteine 57 (Cys57) and Cys146. The protein binds one copper ($Cu^{2+}$) ion for activity and one zinc ion ($Zn^{2+}$) for structure. It has been shown that the depletion of metal ions from SOD1 or the reduction of the disulphide bond can induce misfolding and toxic conformations in SOD1. Several studies have demonstrated that SOD1 can be stabilized and prevented from misfolding and aggregating through the binding of small molecules. One study [8] examined SOD1-ligand interactions using a SOD1 glutathione S-transferase (GST) fusion protein. Using *in silico* docking, they identified about 100 compounds that might fit in the pocket between the SOD1 monomers that could stabilize the homodimer and prevent misfolding. About 15 phenolic compounds were tested to determine if dimers of purified recombinant wtSOD1 or 3 fALS variants were stabilized by these compounds. A number of compounds, with the best being halogenated, inhibited aggregation by increasing the longevity of the dimers of fALS variants. However, these compounds had little effect on wtSOD1 stability. Wright et al. [9] used similar experiments to study SOD1 aggregation and unfolding and found that 5-fluorouridine (FUrd) and isoproterenol bound to SOD1 and stabilized the protein. However, they did not observe any change in the dimer/monomer dynamics when FUrd was mixed with another fALS SOD1 mutant (A4V) nor with a second mutant SOD1 (I113T), both of which have known propensities to aggregate. Wright et al. [9] did observe, using X-ray crystallography, that FUrd interacted with tryptophan 32 (Trp32), through aromatic ring stacking. Isoproterenol also bound to this Trp32 site in the SOD1

(I113T) mutant. A computational study [10] also showed the possibility of a π-stacking inter-actions between FUrd and the Trp32. These results contrast with the suggestion [8] that the dimer interface is the preferred site of therapeutic ligand binding.

It is because of these varying results regarding SOD1 ligand binding that we chose to use nuclear magnetic resonance (NMR) to study SOD1-ligand interactions, in solution, to rule out complications or binding artefacts arising from crystallization. We also chose to study a specially designed monomeric form of SOD1 to ensure that complications arising from monomer-dimer interactions could be ruled out. Given that other studies have also shown that monomerization of dimeric SOD1 is part of a pathway to SOD1 aggregation [11], solution characterization of these ligand interactions would have even greater biological relevance. The monomeric SOD1 protein (called T2M4SOD1) we purified had 6 mutations. Two mutations (T2) corresponding to C6A and C111S are known to slightly increase the thermostability of SOD1 [12]. The remaining 4 mutations (M4) introduce charges at the dimer interface while retaining the overall charge of the wild type subunit [13]. The M4 SOD1 has been shown to be monomeric, up to concentrations of 1 mM, is enzymatically active and binds $Cu^{2+}$ and $Zn^{2+}$ with the correct stoichiometry. Technically, the T2M4SOD1 protein being studied here is a mutant enzyme with 8–35% reduced activity compared to wtSOD1. While reduced activity of SOD1 has been observed with some fALS variants, it is important to emphasize that none of the mutations in our T2M4SOD1 mono-mer correspond to known fALS SOD1 gene mutations. This fact makes T2M4SOD1 a par-ticularly useful choice for performing biophysical studies of wtSOD1-ligand interactions via NMR. This is because the NMR spectra are not only better resolved and simplified, but the folding, stability and ligand binding activity of this engineered SOD1 protein should closely mimic wtSOD1 rather than fALS SOD1.

Previous reports describing the production of monomeric SOD1 for NMR analysis [13–16] provided little detail and had very poor yields. Here we describe a novel, much more efficient protocol for preparing monomeric SOD1 (namely the six-mutation T2M4SOD1) with yields of >15 mg/L of bacterial culture that is readily amenable to NMR-ligand studies. To demonstrate the utility of this monomeric form of SOD1 for NMR studies, we tested its binding to eight pyrimidine compounds which have been shown (or suggested via computa-tional modeling) to bind to SOD1. One of these compounds, FUrd, exhibited line broaden-ing in both $^{1}$H and $^{19}$F spectra when added to a 1 mM solution of T2M4SOD1, suggesting weak binding to the monomer. Using various derivatives of FUrd, particularly uracil (U) and halogenated uracil derivatives (5-fluorouracil, 5-chlorouracil and 5-bromouracil—FU, ClU and BrU), we found enormously increased line broadening (beyond detection) when these compounds were added to 1 mM solutions of T2M4SOD1. These data partially con-firm the X-ray crystallographic findings [9] and recent computational predictions [10]. They have also led to the identification of a new class of uracil analogs that may exhibit even stronger binding than FUrd. However, the aromatic ring stacking interaction between FUrd and Trp32 seen by X-ray could not be seen by our NMR studies due the intermediate exchange dynamics of these ligand-SOD1 interactions. We used computer-based molecular docking calculations as well as molecular dynamics (MD) simulations to explore the bind-ing dynamics of these ligands to SOD1 at an atomic detail and discovered that hydrogen bonding (H-bonding) with aspartate 96 (Asp96) may play an important role in mediating uracil ligand binding. While precise atomic interactions were not as easily detected via NMR with these pyrimidine analogs as hoped, overall, we believe the protocols described here demonstrate that monomeric T2M4SOD1 can be easily prepared and readily used to explore small molecule interactions with SOD1 via NMR.

## Materials and methods

### Materials

Kanamycin sulfate, isopropyl β-D-1-thiogalactopyranoside (IPTG), 100X MEM vitamin mixture (for bacterial culture work), bovine serum albumin (BSA), Bromophenol Blue, 5-bromouracil (BrU), 5-bromouridine (BrUrd), 3-(trimethylsilyl)-1-propanesulfonic acid-$d_6$ sodium salt (DSS-$D_6$), 5-fluorouracil (FU), 5-fluorouridine (FUrd), nitroblue tetrazolium (NBT), riboflavin, uracil (U) and uridine (Urd) were purchased from Sigma (Oakville, Canada). 5-chlorouracil (ClU) was obtained from Ark Pharm Incorporated (Arlington Heights, USA). The trifluridine (F3TDR) was purchased from Oxchem Corporations (Wood Dale, USA). The BLUelf prestained protein ladder was purchased from FroggaBio (Concord, Canada). The Luria Broth (LB, Invitrogen), Coomassie G-250, HPLC grade $H_2O$ and the Spectra/Por 6–8 kDa molecular weight cut-off (MWCO) dialysis tubing were purchased from Fisher Scientific (Ottawa, Canada). The 3.5 kDa MWCO Snakeskin dialysis tubing was purchased from Thermo Fisher Scientific (Whitby, Canada). The $^{15}NH_4Cl$ was purchased from Cambridge Isotope Laboratories (Andover, USA). The MAXYMum Recovery 1.5 mL centrifuge tubes were purchased from Corning (Fairport, USA). The 3 mM SampleJet NMR tubes were obtained from Bruker (Milton, Canada).

### Cloning and expression of the T2M4SOD1 monomer

The design of the M4 mutant SOD1 monomer was previously described [13] and has 4 mutations that introduce charges at the dimer interface (F50E, G51E, V148K, and I151K). Two mutations (T2) increased the thermostability of SOD1, C6Aand C111S, as previously described [12,17]. We decided to combine both sets of mutations to produce a more thermally stable and potentially more easily foldable monomeric SOD1 protein: T2M4SOD1. To further improve the protein yield and simplify purification, the leader peptide from the bacteriocuprein from *Photobacterium leiognathi* [18] was fused to the N terminus of T2M4SOD1. The synthetic gene was engineered to have 5'*Nde*I and 3'*Bam*HI restriction sites and was inserted into *Nde*I-*Bam*HI digested pET24a (+) which removed the T7-Tag from the vector (Fig 1A). The gene synthesis with codon optimization for expression in *Escherichia coli* (*E. coli*) and express cloning (including insertion into the pET24a (+) plasmid), was performed and provided by Genscript (Piscataway, USA). The final DNA sequence for T2M4SOD1 is shown in Fig 1B. Upon receipt of the clone, the plasmid was transformed into *E. coli* BL21 (DE3) competent cells via standard heat shock [19] and colonies were grown on LB with 30 μg/ml kanamycin agar plates. The DNA sequence was confirmed by Sanger sequencing (Molecular Biology Service Unit, Department of Biological Sciences, University of Alberta).

To optimize the expression of unlabeled T2M4SOD1, we grew the cells containing the T2M4SOD1 expression plasmid under a variety of conditions (different temperatures, different periods of time, different concentrations of IPTG and different concentrations of metals). In particular, cells were grown in rich LB at room temperature (RT; 21–22˚C), 28˚C and 37˚C with 0.2, 0.4, 0.6, 0.8 and 1 mM IPTG and/or 100 μM $CuSO_4$ and 30 μM $ZnSO_4$. A single colony was inoculated into 10 mL of LB with 30 μg/mL kanamycin in a 50 mL sterile centrifuge tube and grown overnight at 37˚C, 250 rpm. Then 1.25 mL of LB of the overnight starter culture was inoculated into 125 mL of LB plus 30 μg/mL kanamycin and grown at 37˚C, 250 rpm until an $OD_{600}$ between 0.6–0.8 was reached. Cells were then induced with different concentrations of IPTG and then split into 6 x 20 mL aliquots in 125 mL flasks. After IPTG induction, the cultures were grown with or without added 100 μM $CuSO_4$ and 30 μM $ZnSO_4$ at the indicated temperatures for 2 hrs, 4 hrs or overnight. 1 mL aliquots of culture were removed before

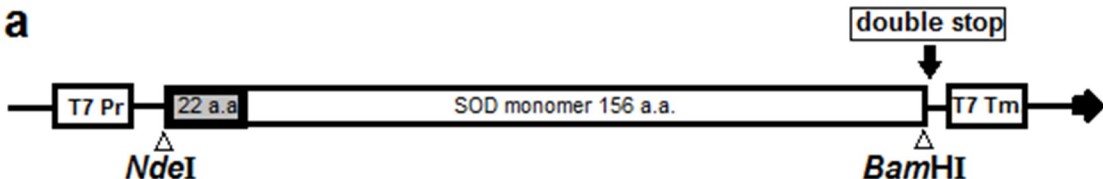

**Fig 1.** The schematic structure (a) and nucleotide sequence (b) of the superoxide dismutase (SOD1) monomer (T2M4) inserted into pET24a (+) plasmid. The gene encodes the bacteriocuprein leader peptide (22 residues long) from Photobacterium leiognathi and the 153 amino acids of the human SOD1 monomer (M4 SOD1- F50E, G51E, V148K, I151K), that is thermostable (T2- C6A, C111S). Transcription is driven by the T7 RNA polymerase. T7 Pr -T7 promoter; T7 Tm -T7 terminator. **(b)** The nucleotide sequence of the synthetic gene of the SOD1 monomer contains the 22 amino acid leader peptide (dashed line under nucleotide sequence**)** and two stop codons (double underlined). The first amino acid of T2M4SOD1, alanine, is underlined with a wavy line. The 5' NdeI and the 3' BamHI sequences are underlined.

and after 2 hrs or 4 hrs or after overnight growth with IPTG. Similar experiments were repeated inducing T2M4SOD1 expression in LB medium or minimal M9 medium with 0.1, 0.2, 0.4 and 0.8 mM IPTG, with or without metals, after 2 hrs, 4 hrs and after overnight growth. The M9 medium contained 6.8 g $Na_2HPO_4$, anhydrous; 3 g $KH_2PO_4$; 0.5 g NaCl; 1 g of $^{15}NH_4Cl$ or $NH_4Cl$ in 1 L of Milli-Q water.

After the optimized protocols were determined for both the labeled and unlabeled growth and induction conditions for T2M4SOD1, we prepared the protein samples as follows. For purification of unlabelled T2M4SOD1 in rich media, a single colony was inoculated into 25 mL of LB with 30 μg/mL kanamycin and grown at 37˚C, overnight with 250 rpm shaking. A 1 L solution of LB with 30 μg/mL kanamycin was inoculated with 10 mL of the overnight starter culture and grown at 37˚C, 250 rpm until an $OD_{600}$ between 0.6–1.0 was reached. Cells were induced with 0.2 mM IPTG and 100 μM $CuSO_4$ and 30 μM $ZnSO_4$ and grown at RT, overnight (about 20 hours), with shaking at 200 rpm. Cells were harvested by centrifugation at 3500 x *g*, 4˚C for 20 min. The supernatant was decanted off and the cell pellets were subjected to osmotic shock to release the contents of the periplasm.

For the preparation of uniformly labelled $^{15}N$-T2M4SOD1 or unlabelled T2M4SOD1 in minimal media, the protein was purified from cultures grown in 1 L of M9 medium supplemented with a commercial vitamin mix (0.1% MEM vitamins), 2 mM $MgSO_4$, 200 μM $CaCl_2$ and 4 g of glucose. 20 mL of an overnight culture grown in LB medium with 30 μg/mL kanamycin was inoculated into 1 L of M9 media containing 30 μg/mL kanamycin. Note that no zinc or copper salts were added to the M9 media. The cells were grown to an $OD_{600}$ of about 0.6–0.8, induced with 0.2 mM IPTG and grown overnight at RT, 250 rpm. The next day, cells were harvested by centrifugation as described above.

## Purification of T2M4SOD1

The T2M4SOD1 protein was purified by osmotic shock, ammonium sulfate precipitation and anion-exchange chromatography using a modification of the method previously described [15]. The pelleted cells were resuspended in a solution of 50 mL of 20% sucrose in 10 mM Tris, pH 7.7. This volume of sucrose solution approximated the ratio of equivalent volumes reported by [20] with 1 mL of 1 x $10^8$ cells resuspended in 0.15 mL of sucrose solution. Then 500 mM EDTA, pH 8, was added to a final concentration of 16.6 mM EDTA, and the sucrose-EDTA mixed cells were incubated on ice for 1 hr, without stirring. The cells were pelleted at 16,000 x $g$ for 20 min at 4˚C. The sucrose-rich supernatant was decanted off and 200 mM $CuSO_4$ and $ZnSO_4$ were added to the sucrose fraction to a concentration of 33 mM (since the 16.6 mM EDTA in the previous step would have stripped the metals from the T2M4SOD1). The sucrose fraction was kept and stored at 4˚C. The cell pellet was resuspended with 25 mL of 100 μM $CuSO_4$ (approximately half of the final volume of the sucrose solution) and kept on ice for 30 min to release the contents of the periplasm by osmotic shock. The cells were pelleted again at 16,000 x $g$ for 20 min at 4˚C. The supernatant, containing the periplasmic contents, were decanted off and combined with the previously stored sucrose fraction. The combined pool was heated to 65˚C in a water bath for 2 hrs and then immediately cooled in ice water and stored overnight at 4˚C. Heat labile proteins were pelleted at 16,000 x $g$ for 20 min, 4˚C and the supernatant carefully decanted into a graduated cylinder to determine the volume of the sucrose/periplasm pool. Proteins from the supernatant were precipitated with 50% (0.291 g/mL), then 65% (with the addition of a further 0.092 g/mL) ammonium sulfate, $(NH_4)_2SO_4$, stirring for 1 hr at 4˚C (in a cold room). Ammonium sulphate precipitation was done by slowly adding solid $(NH_4)_2SO_4$ to the ice-water chilled sucrose/periplasm pool. After each $(NH_4)_2SO_4$ addition (first the 50% cut, then the 65% cut), precipitated proteins were pelleted 12,000 x $g$ for 20 min at 4˚C and the volume of the supernatant was determined to accurately add the next amount of $(NH_4)_2SO_4$. The 65% $(NH_4)_2SO_4$ supernatant was then increased to 100% $(NH_4)_2SO_4$ with the addition of 0.244 g/mL of solid $(NH_4)_2SO_4$ and stirred overnight. The precipitated proteins from the 100% $(NH_4)_2SO_4$ cut were pelleted as described previously. The 65% $(NH_4)_2SO_4$ supernatant, the 100% $(NH_4)_2SO_4$ supernatant and the pellet were tested to determine the amount of T2M4SOD1 present in these $(NH_4)_2SO_4$ cuts. For purification, the proteins in the 50% $(NH_4)_2SO_4$ supernatant only were dialyzed at 4˚C in Spectra/Por 6–8 kDa MWCO dialysis tubing with 3 changes (for 24 hrs each) of 4 L of a dialysis solution consisting of 2 mM potassium phosphate buffer, 100 μM EDTA, pH 8. The dialyzed protein solution was then loaded onto a 13 x 100 mm DEAE-Sephacel column (packed at RT), equilibrated with dialysis buffer, at a flow rate of 1 mL/min using an AKTA Prime chromatography system (Amersham Biosciences). After loading the protein, the T2M4SOD1 was eluted from the ion exchange column with a gradient of 2–100 mM potassium phosphate buffer, 100 μM EDTA, pH 8 over 5 column volumes with a flow rate of 2 mL/min. Proteins contained in the eluted peaks were analyzed for the presence of T2M4SOD1 on 12% Tricine- SDS-PAGE and stained with Coomassie G-250 [21]. The fractions containing T2M4SOD1 (which typically eluted with the 3rd column volume) were pooled, dialyzed against 1L of 1 mM $CuSO_4$ in 20 mM Tris, pH 8 at 4˚C for 24 hrs, then 2 changes of 4.5 L of Milli-Q $H_2O$ (18 mΩ) for 12 hrs each and lyophilized. The predicted molecular weight of the purified T2M4SOD1, with the leader peptide removed (153 amino acids, 15854.49 Da) was confirmed using liquid chromatography/mass spectrometry (LC/MS) as performed by the Department of Chemistry Mass Spectrometry Laboratory, University of Alberta.

### In-gel superoxide dismutase enzymatic assay

The purified T2M4SOD1 protein was examined for activity using the in-gel enzymatic assay originally described by [22]. The purified T2M4SOD1 was mixed with 5X sample buffer (50% glycerol, 0.75 mM Tris, pH 8.8, 0.5% Bromophenol Blue) and loaded onto a 10% native PAGE gel with a 0.375 mM Tris, pH 8.8 buffer, without a stacking gel. BSA was used as a MW standard. The T2M4SOD1 has a predicted isoelectric point of 5.72 (ExPASy ProtParam tool [23]) and would be negatively charged at pH 8.8. The gel apparatus was kept cold and run for 2 hours at 100V with normal polarity. The gel was soaked with 1 mg riboflavin, 2.5 mg NBT in 10 mL Milli-Q $H_2O$ for 15 minutes, in the dark, at RT, with gentle rocking. After the soaking, the gel was washed briefly with Milli-Q $H_2O$ and then incubated 15 min in the dark with 0.1% TEMED in 10 mL Milli-Q $H_2O$. The gel was exposed to light with gentle rocking, while the color developed (about 10 minutes), and then washed with Milli-Q $H_2O$.

### NMR ligand binding studies with T2M4SOD1 and $^{15}$N-T2M4SOD1

All NMR samples were prepared with a 20 mM potassium phosphate buffer, pH 7.4, 10% $D_2O$ with 1 mM DSS-$D_6$ as chemical shift reference. Samples were prepared to 200 μL final volume using 1.5 mL MAXYMum recovery microfuge tubes and then loaded into 3 mm NMR tubes. All tested ligands are listed in Table 1. The T2M4SOD1, 380 μM, alone, FUrd (45 μM, 450 μM and 4.5 mM) alone or 380 μM T2M4SOD1 mixed with 45 μM, 450 μM and 4.5 mM FUrd (0.11X, 1.1X or 11X protein to ligand, respectively) were examined. We tested 200 μM, 2 mM or 10 mM (1X, 10X and 50X, respectively) of U, BrU, ClU, FU, BrUrd or F3TDR alone or mixed with 200 μM T2M4SOD1. The FUrd and T2M4SOD1 samples were pooled, dialyzed, using 3.5 kDa MWCO Snakeskin dialysis tubing, against 10 L total, at 4°C, of Milli-Q $H_2O$ for 51 hours (3 hrs in 4 L, 24 hrs in 2$^{nd}$ 4 L and finally 24 hrs in another 2 L) and then confirmed to be free of 5-FUrd by NMR. $^{1}$H-NMR spectra were collected from 125 μM T2M4SOD1, alone, 94.6 μM, 946 μM or 4.7 mM Urd alone or 1X, 10X or 50X Urd mixed with 125 μM T2M4SOD1. NMR spectra were collected from samples with 440 μM $^{15}$N-T2M4SOD1, alone, or samples containing 4.4 mM Urd, U or FU mixed with 440 μM $^{15}$N-T2M4SOD1. The U, BrU, ClU and FU were first solubilized with small amounts of 5N NaOH and then adjusted to pH 7–8 with molar equivalents of HCl. The pH was confirmed to be between 7–8 using pH paper.

### NMR spectra collection

$^{1}$H-, $^{15}$N- and $^{19}$F-NMR spectra were collected using a 700 MHz Bruker NMR spectrometer equipped with a 5 mm triple resonance cryoprobe. The 1D-$^{1}$H-NMR spectra were recorded using the *zgesgppe* pulse sequence to suppress the water signal [24]. Spectra were recorded

**Table 1. Compounds and their substitution pattern tested for binding with T2M4SOD1 by NMR spectroscopy.**

| Compound | NMR Observations | Interaction with SOD |
|---|---|---|
| **FUrd 5-fluorouridine** | Broad signals, in particular $^{19}$F (broader than F-Uracil) | yes |
| **Urd uridine** | Similar shifts, no line broadening | no |
| **BrUrd 5-bromouridine** | Similar shift, limited broadening | very weak |
| **F3TDR trifluridine** | Similar shift, limited broadening | very weak |
| **FU 5-fluorouracil** | Broad $^{19}$F, very broad $^{1}$H, even at 50-fold excess (loss of signal) | yes |
| **U uracil** | Broadened signals | yes |
| **BrU 5-bromouracil** | Very broad signals, even at 50-fold excess (loss of signal) | yes |
| **ClU 5-chlorouracil** | Very broad signals, even at 50-fold excess (loss of signal) | yes |

with a spectral width of 12 ppm using 128 transients, an acquisition time of 4 s and a relaxation delay of 2 s. The 1D-[19]F-NMR was recorded using a simple 90° pulse sequence with a spectral width of 40 ppm using 128 transients, an acquisition time of 2.5 s and a relaxation delay of 2 s. The [1]H-[15]N-spectra were recorded using the *hsqcetfpf3gpsi* sequence with standard parameters (1024 points along F2, 80 points along F1, 32 scans, with a spectral width of 12 and 30 ppm, respectively). The [1]H dimension was referenced directly to DSS and the [15]N dimension was referenced indirectly by their gyromagnetic ratios following the recommendations of Wishart et al. [25]. Spectra from T2M4SOD1 at protein concentrations ranging from 125–440 μM, alone or mixed with ligands were acquired and compared to the [1]H-NMR and/or [19]F-NMR spectra obtained with the ligands alone at the previously recorded concentration to eliminate possible stacking effects of the ligand. In addition, saturation transfer difference (STD) spectra were recorded with an *esgp* pulse sequence to suppress the water [26]. The STD spectra were recorded with a 12 ppm spectral width using 128 transients, with an acquisition time of 4 s, a recycle delay of 5 s and an irradiation time of 2 s with a Gaussian-shaped pulse for selective saturation of the protein resonances. The on-resonance irradiation frequency was set to 580 Hz (methyl-groups), 1062 Hz (methylene-groups) and 7175 Hz (indole-proton of the Trp side chain) and off-resonance at 35000 Hz. Difference-spectra of off- and on-resonance spectra were calculated and used to determine possible STD effects.

### Molecular docking calculations

All molecular docking simulations used the AutoDock4 software package [27]. The receptor structure was adopted from the X-ray crystallographic structure of the human SOD1 I113T mutant complexed with 5-FUrd (PDB ID 4A7S; [9]). To locate binding sites, the entire receptor surface was subjected to a systematic grid search using a population size of 100 and 150 genetic algorithms (GA) runs. All the other docking parameters used the default settings provided with the software. Docked complexes were clustered using 2.5 Å geometry-based cut off. As we observed binding of all the ligand sites close to the Trp32 residue, we ran additional docking calculations using a grid of 30 Å$^3$ centered on the Trp32 residue. Complexes obtained from these docking runs were used for further analysis.

### MD simulations

For all the MD simulations, we started from the docked complexes obtained from the above docking simulations, selecting docked conformations closest to the Trp32 residue. The ligands were parameterized using the CHARMM force field parameters as provided by the SwissParam server [28]. The protein molecule was treated with the CHARMM36 force field parameters [29]. The protein-ligand complexes were immersed in a standard dodecahedral TIP3P water box and neutralized using Na+ ions. The system was energy minimized by subjecting it to 1 ns each of NVT (constant Number, Volume and Temperature) and NPT (constant Number, Pressure and Temperature) equilibrations at 298.15 K and 1 bar pressure using a standard Berendsen thermostat under periodic boundary conditions. The 250 ns production run was carried out using the equilibrated structures. All the trajectory manipulations were done using the standard tools supplied with the GROMACS MD simulation package [30].

### Electronic structure calculations

All the electronic structure calculations used the Gaussian16 quantum chemistry program [31]. The binding site of the ligands (residues within 5 Å of the ligand molecules) were generated from the docking simulations. Two different types of calculations were performed on the structures containing the Trp32 site from the receptor. First, H-atoms identified in the binding

poses from the docking simulations were optimized using the HF/3-21G* level of theory [32,33]. These structures were further used for NMR isotropic shielding tensor calculations at the density functional theory (DFT) based M06-2X/6-311++G** level of theory using the gauge independent atomic orbital (GIAO) method [34–36]. In the second set of calculations, the amino acid backbone of the binding site was frozen. The ligand and side chains were optimized at the M06-2X level using the 6-31G** basis set. We chose the dispersion-corrected density functional M06-2X, a method that incorporates π-stacking, because previous reports indicated that pyrimidines can π-stack with the indole side chain of Trp32.

## Results

### Expression of T2M4SOD1

The M4SOD1 monomer was previously described and expressed in the *E. coli* strain MC1061 from a pBR322 plasmid driven by a tac promoter [13]. Our T2M4SOD1 construct contained the leader peptide from bacteriocuprein from *Photobacterium leiognathi*, which sorted the T2M4SOD1 gene product to the periplasm where the leader peptide was removed. Expression of this SOD1 construct, inserted into the *Nde*I-*Bam*HI of pET24a(+), was driven by the T7 promoter and was placed into the protease depleted *E. coli* strain BL-21 (DE3; Fig 1A). The construct also contained 2 tandem stop codons to ensure that these transcripts would be terminated precisely at the end of the SOD1 gene (Fig 1A). The codons of the nucleotide sequence of T2M4SOD1 including the N-terminal 22 amino acids of the bacteriocuprein leader peptide (Fig 1B) were codon optimized for expression in *E. coli* and further optimized by removing any possible secondary RNA structures (a service provided by Genscript). The DNA was isolated from transformed cells and was sequenced at the Molecular Biology Service Unit, Department of Biological Sciences, University of Alberta. The ExPASy translate function [23] was used to confirm that the gene construct contained the published amino acids sequence of the T2M4SOD1.

When Banci et al. [13] described their purification of the M4SOD1 monomer, little experimental or methodological detail was provided. We also note that no other group has since been able to produce the monomeric SOD1 protein. Using the minimal information from the Banci et al. publications [13,14] we attempted to reproduce their work with our T2M4SOD1 construct, but without success. As a result, we undertook an effort to develop a different protocol and to optimize our T2M4SOD1 expression in both rich medium as well as minimal medium as the latter is required for isotopic labelling of proteins used in NMR studies. We examined the expression of leader peptide-T2M4SOD1 (lp-T2M4SOD1) at different temperatures, increasing amounts of IPTG and with/without $Cu^{2+}$ and $Zn^{2+}$ ions as we did not know if an enzymatically active T2M4SOD1 would properly fold if these metal ions were not present in the induced culture media. We also wanted to confirm that the leader peptide was removed and that the leader-peptide-free T2M4SOD1 only sorted to the periplasm. When the lp-T2M4SOD1 was grown in rich LB medium with or without metals and expression induced with 1mM IPTG at RT (Fig 2A, lanes 1–4), 28°C and 37°C (Fig 2A, lanes 5–7), after 2 and 4 hours induction, both the lp-T2M4SOD1 and T2M4SOD1 were detected by Coomassie-staining of proteins separated by Tricine-SDS-PAGE. Less T2M4SOD1 was only seen in the LB-cultures grown overnight at 37°C (Fig 2A, lane 7). The presence of metals did not affect expression of the lp-T2M4SOD1 and T2M4SOD1. Growth at RT and 28°C produced better overnight expression compared to growth at 37°C (Fig 2A, compare lane 4 to lane 7). Our results suggested that T2M4SOD1 did have the leader peptide removed but that the lp-T2M4SOD1 was also present in whole cells.

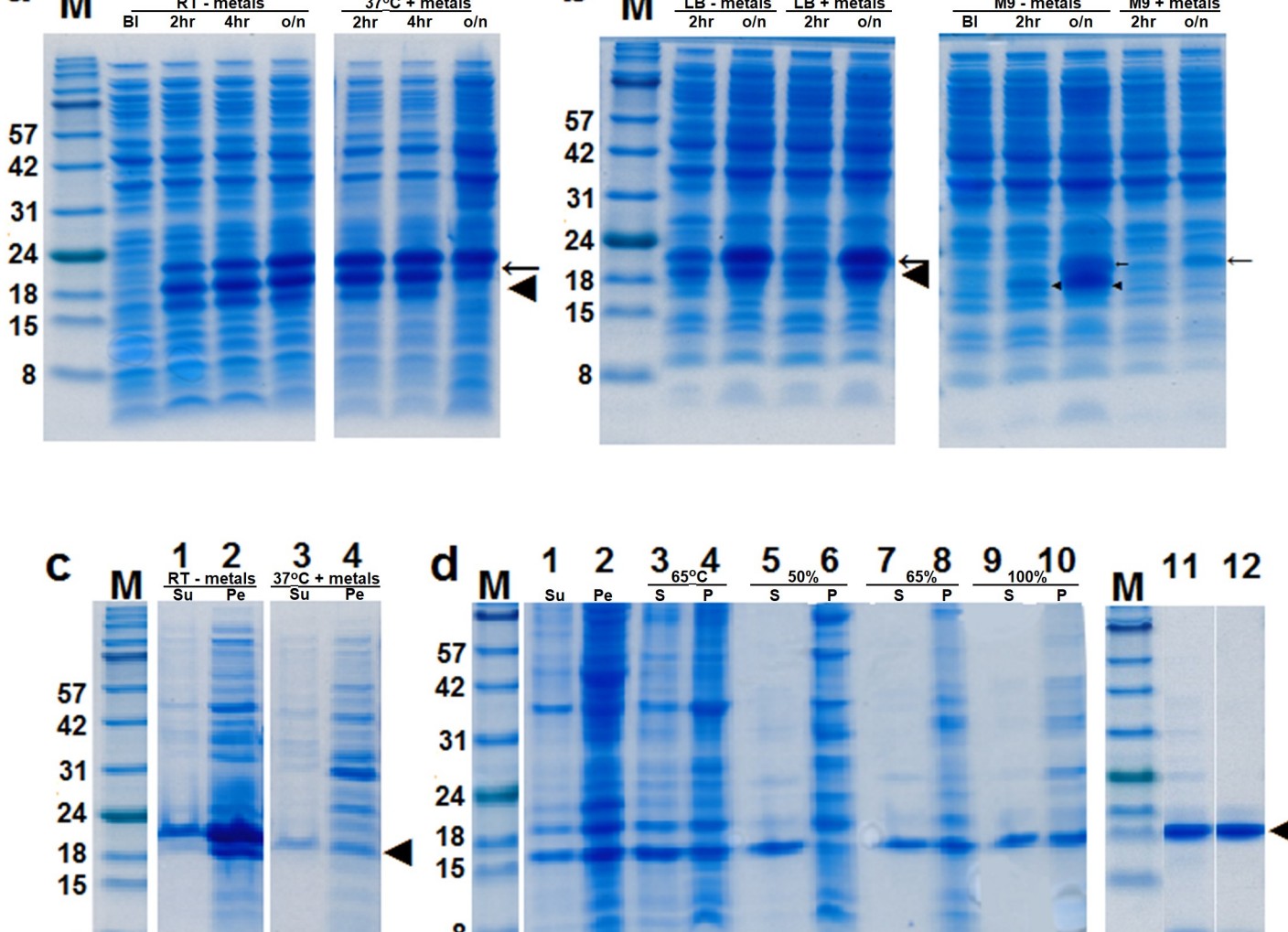

**Fig 2. Expression of leader peptide-T2M4SOD1 in rich and minimal media, its periplasmic extraction and purification of $^{15}$N-T2M4SOD1. (a)** Aliquots from E. coli BL-21 (DE3), grown in LB medium at RT (a, lanes 2–4) and 37˚C (a, lanes 5–7), were removed before (BI, a, lanes 1) and after 2 hr, 4 hr and after overnight induction with 1mM IPTG without (-) metals (lanes 2–4, respectively) and with (+) metals (lanes 5–7, respectively). The leader peptide-containing T2M4SOD1 (←) and the leader peptide-free T2M4SOD1 (◄) were found in whole cells separated by Tricine-SDS-PAGE gel and Coomassie-stained, with less T2M4SOD1 seen after overnight expression at 37˚C with metals. **(b)** Aliquots from E. coli BL-21 (DE3), grown in LB (lanes 1–4) or M9 media (lanes 5–9) at RT before induction (BI, lane 5) and after induction without metals and with 1mM IPTG (lanes 1–2, 6–7) and with metals and 0.1 mM IPTG (lanes 3–4, 8–9), were removed after 2 hr growth (lanes 1, 3, 6, 8) and after overnight growth (lanes 2, 4, 7, 9). With more inducer, more LPT2M4SOD1(←) and T2M4SOD1 (◄) were detected after 2 hours in LB but the same amount of expression was seen after overnight growth with 0.1 mM or 1 mM IPTG, with or without metals. In contrast, when metals were added to M9-grown cultures, only the LP-T2M4SOD1 (←) was detected after 2 hours with slightly more seen after overnight growth. The T2M4SOD 1 (◄) was only seen after 2 hours growth without metals and was more abundant than the LP-T2M4SOD1 (←) in M9-grown cultures after overnight growth. **(c)** Outer membranes and periplasm were extracted from LB-grown cultures grown at RT without metals (lanes 1–2) and grown at 37˚C with metals (lanes 3–4) using sucrose/EDTA (Su, lanes 1 and 3) and osmotic shock with 100 μM CuSO4 (Pe-periplasm, lanes 2 and 4). The osmotic shock fraction contained more T2M4 SOD1 than the outer membrane fraction. Much less Coomassie-stained T2M4SOD1 was seen after growth at 37˚C with metals than at RT growth without metals. **(d)** Purification of $^{15}$N-T2M4SOD1 from M9 cultures grown overnight at RT with 0.2 mM IPTG. The outer membrane (Su-lane 1) and periplasmic fractions (Pe-lane 2) were pooled, heated for 2 hr at 65˚C and pelleted. The supernatant (S-lane 3) and pelleted heat-labile proteins (P-lane 4) both contained T2M4SOD1. The proteins from the heated supernatant were separated by precipitation with 50%, 65% and 100% NH4SO4. The pelleted (P), precipitated proteins are shown in lanes 6, 8 and 10 while the proteins in the supernatant (S) are shown in lanes 5, 7 and 9. The proteins in the 50% supernatant (lane 11) were separated using DEAE-Sepahacel with T2M4SOD1 eluting with 50–60 mM potassium phosphate (lane 12). The prestained protein molecular weight standards (M) are in kDa.

To determine if more of the leader-peptide-free T2M4SOD1 would be made when less IPTG was added to the LB-grown cells, we induced the lp-T2M4SOD1 cells with 0.1, 0.2, 0.4, 0.8 and 1 mM IPTG at RT (21–22˚C) with or without metals. With increasing concentrations of IPTG, with or without metals, more T2M4SOD1 was detected at 2 hours with 1 mM IPTG (Fig 2B, compare lanes 1 and 3) but about equivalent amounts were seen with all IPTG concentrations after overnight growth (Fig 2B, compare lanes 2 and 4). We decided to induce cells with less IPTG (0.2 mM) to slow down transcription and translation of the T2M4SOD1 protein.

To generate isotopically labelled protein, cultures must be grown in minimal medium, free of any source of unlabeled amino acids. While some proteins may be well expressed in minimal medium, such as our water soluble Aβeta$_{42}$ peptide variant [37], not all proteins are well expressed in minimal compared to rich media. Therefore, we needed to optimize conditions for expression of T2M4SOD1 in minimal medium. We tested several induction conditions with 0.1, 0.2, 0.4, 0.8 and 1 mM IPTG at RT (21–22˚C) with $Cu^{2+}$ and $Zn^{2+}$ as well as with 1 mM IPTG and without $Cu^{2+}$ and $Zn^{2+}$ as a control. We observed reduced expression of T2M4SOD1 in minimal medium without $Cu^{2+}$ and $Zn^{2+}$ at 2 hours compared to rich medium (Fig 2B, compare lane 6 and 1) and more leader peptide-less T2M4SOD1 after overnight growth at RT with 1 mM IPTG (Fig 2B, compare lanes 7 to 2). However, when $Cu^{2+}$ and $Zn^{2+}$ were added and cultures were induced with 0.1 mM to 1 mM IPTG, some lp-T2M4SOD1 was detected but no T2M4SOD1 was seen (Fig 2B, lanes 8–9 with 0.1 mM IPTG). Moreover, the cells did not grow well. Therefore, we determined that isotopically labelled T2M4SOD1 could only be expressed and purified in cultures grown in M9 media without the addition of $Cu^{2+}$ and $Zn^{2+}$.

## Purification and characterization of T2M4SOD1

We purified T2M4SOD1 from the periplasm of cultures grown in LB medium with $Cu^{2+}$ and $Zn^{2+}$ or minimal medium without $Cu^{2+}$ and $Zn^{2+}$. Both were induced with 0.2 mM IPTG at RT and grown overnight such that the T2M4SOD1 could be fractionated and largely purified the following day from freshly pelleted cells. While protocols have been described for the column-based purification of monomeric SOD1 [13,15], the details were brief and contradictory. As a result, we developed our own purification protocol.

As seen in Fig 2B, in whole cells, two proteins bands were detected after induction of expression with IPTG in both rich LB and minimal M9 media, respectively. This suggested that both T2M4SOD1 fused to the leader peptide as well as T2M4SOD1 alone were present in the cells. Only periplasmic proteins should be isolated since the leader peptide-fused-T2M4SOD1 may yet be present within the cytoplasm. Since the bacterial cytoplasm contains about tenfold more proteins than the periplasm [38], we decided to avoid freezing the cells in order to prevent the potential dilution of the periplasmic pool through freeze-thaw damage of the cytoplasmic membrane. To isolate the periplasmic proteins, we first released loosely bound membrane proteins from whole cells by resuspending freshly pelleted cells with sucrose (3 mL for 20 mL of culture and 50 mL for 1 litre of culture) and then adding EDTA (16 mM) for 1 hour on ice, without stirring. After the cells were pelleted, they were resuspended with 100 μM copper sulfate (half the volume of the final sucrose volume) for 30 min on ice, without stirring, to osmotically shock the cells and release the contents of the periplasm. The T2M4SOD1 was detected on Coomassie-stained Tricine-SDS-PAGE gels in both the sucrose and the periplasmic fractions with much more T2M4SOD1 released with osmotic shock from LB cells grown overnight at RT (Fig 2C, lanes 1 and 2), compared to cell grown overnight at 37˚C (Fig 2C, lanes 3 and 4). The same pattern of protein release was also observed from cells grown in

minimal medium with $^{15}NH_4Cl$ (Fig 2D, lanes 1 and 2, respectively). Because the T2M4SOD1 was present in both the sucrose and osmotic shock fractions, both fractions were combined to maximize yields. This pool was then heated at 65 °C for 2 hours to denature and precipitate the heat-labile proteins. Because the T2M4SOD1 is a thermostable protein, it was expected to be unaffected by heating. After centrifugation, more T2M4SOD1 was seen in the supernatant than the pellet; and the pellet also contained several heat labile proteins, including small amounts of T2M4SOD1 (Fig 2D, lanes 3 and 4, supernatant and pelleted proteins, respectively). When the proteins of the heated supernatant were precipitated with 50% $(NH_4)_2SO_4$, the T2M4SOD1 was found in the supernatant and not in the pelleted proteins (Fig 2D lane 5 and 6, respectively). However, when this 50% $(NH_4)_2SO_4$ supernatant was precipitated with 65% $(NH_4)_2SO_4$ and then 100% $(NH_4)_2SO_4$, the T2M4SOD1 was detected in both the supernatant and pellet (Fig 2D lanes 7–10). To minimize this loss, the T2M4SOD1 was purified from the 50% $(NH_4)_2SO_4$ supernatant only, which was dialyzed against water, then potassium phosphate (Fig 2D, lane 11) and finally separated by anion exchange chromatography. The original SOD1 purification protocol [15] used DE52 as the anion exchanger but this resin is no longer available. We purified the T2M4SOD1 using DEAE-Sephacel and a gradient of 2.5–100 mM potassium phosphate. The T2M4SOD1 eluted around 55–65 mM potassium phosphate (Fig 2D, lane 12).

To confirm that the purified T2M4SOD1 did not have a leader peptide, the DEAE-Sephacel-purified protein was analyzed by LC-MS at the Mass Spectrometry Facility, Department of Chemistry at the University of Alberta. The predicted molecular weight of T2M4SOD1 is 15854.49 Da and with the leader peptide attached is 18183.29 Da. By LC-MS analysis, the T2M4SOD1 had a molecular weight of 15853 Da (Fig 3A), which is remarkably close to the predicted molecular weight. The intra disulfide bond in T2M4SOD1 may account for the 1.5 Da difference between the predicted and the actual weights. Based on the LC-MS data, the leader peptide was removed when the T2M4SOD1 sorted to the periplasm.

We confirmed that the T2M4SOD1 was active using an in-gel enzymatic assay (Fig 3B, right gel). The gel first was immersed with riboflavin and NBT, and then lastly TEMED. When the gel was exposed to light, riboflavin is oxidized while the NBT is reduced to a purple formazan. If the SOD1 protein is active, it should prevent the oxidation of riboflavin and the concomitant reduction of NBT to a purple color. Clear, colorless bands were detected suggesting that the T2M4SOD1 was active. This confirmed that the T2M4SOD1 had superoxide dismutase activity. The clear band of SOD1 activity corresponded to a single band in the Coomassie stained gel (Fig 3B, left gel). A single gel was used to separate T2M4SOD1 that was stained with Coomassie G-250 and tested for superoxide dismutase activity using riboflavin/NBT and TEMED. Compared to BSA as a standard (66 kDa), the T2M4SOD1 migrated further than BSA as a single band in the Tris Native PAGE Coomassie-stained gel. Addition of 1 mM $Cu^{2+}$ and 1 mM $Zn^{2+}$ (+) did not alter the migration or in-gel activity of T2M4SOD1 suggesting that the T2M4SOD1 contained the necessary metal ions and was active (Fig 3B, left and right gels). Because the purified T2M4SOD1 was active in the in-gel enzymatic assay, we infer that it was properly metallated with copper and zinc. A 700 MHz $^1$H-NMR spectrum was collected of T2M4SOD1 (Fig 3C) to determine if the protein was properly folded. As disperse NMR peaks were observed around 7.5–8.5 ppm (corresponding to hydrogen-bonded amide shifts) and -1 to +1 ppm (corresponding to ring-current shifted methyl groups), this clearly indicated that the protein was folded.

## T2M4SOD1-ligand studies

We first established that the interactions with T2M4SOD1 could be monitored by spectral changes in the ligand using NMR spectroscopy. The pyrimidine compound, FUrd, as reported

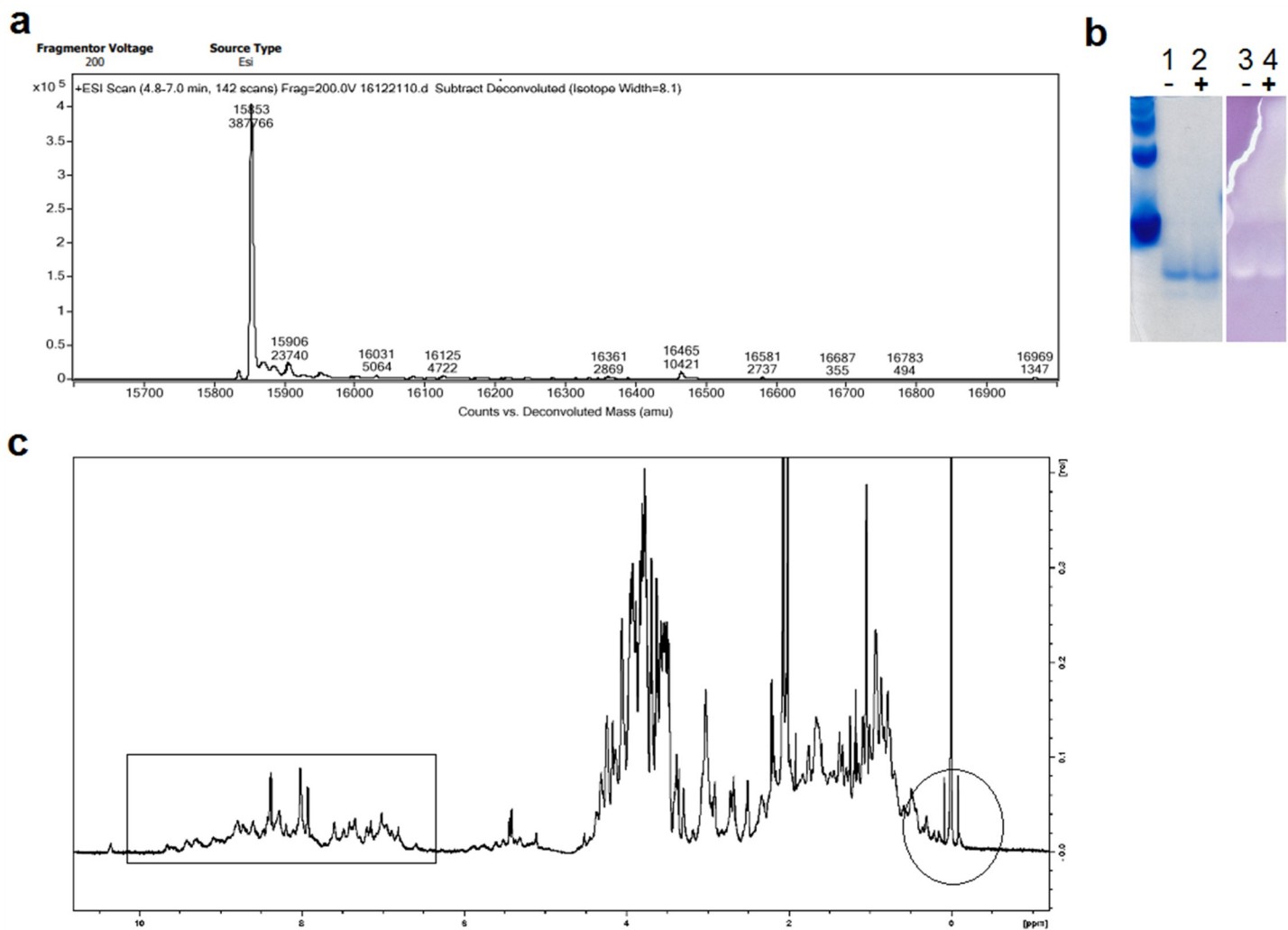

**Fig 3. Characterization of T2M4SOD1 by LC/MS, in-gel enzymatic assay and NMR. (a)** T2M4SOD1, separated by LC/MS, has the Expasy-predicted molecular weight, less 1.5 Da. **(b)** Tris-Native PAGE separation of T2M4SOD1 (with BSA as a standard), stained with Coomassie (left) and by in-gel enzymatic assay using riboflavin and NBT (right). The right gel was first immersed with riboflavin and NBT, then TEMED. When the gel was exposed to light, riboflavin is oxidized and NBT is reduced to a purple formazan. Clear colorless bands were detected, suggesting that the T2M4SOD1 was active, preventing the oxidation of riboflavin and preventing the concomitant reduction of NBT. These bands corresponded to a single band in the Coomassie stained gel (left). Addition of 1mM $Cu^{2+}$ and $Zn^{2+}$ (+) did not alter the migration or in-gel activity of T2M4SOD1. **(c)** [1]H-NMR spectra of 0.4mM T2M4SOD1 in potassium phosphate buffer. The monomer is folded as peaks were observed around 7.5 ppm and -1 to +1ppm, indicative of hydrogen-bonded amide and ring-current shifted methyl groups (in rectangle and circle, respectively).

by X-ray crystallographic studies, was shown to interact with the Trp32 of a I113T mutant SOD1 [9]. We first examined FUrd alone and with T2M4SOD1 to confirm that the claimed X-ray T2M4SOD1 protein-FUrd interaction was detectable via NMR. Studying FUrd alone, both [1]H-NMR and [19]F-NMR chemical shifts occurred suggesting that FUrd tended to stack on itself (Fig 4A, left-[1]H spectrum; 4b-left, [19]F spectrum, compare 1X to 10X). Nevertheless, in the presence of T2M4SOD1, significant line broadening was observed (15.2 Hz and 8.3 Hz for 1X and 10X ligand to protein, respectively, Table 2) indicating binding (Fig 4A and 4B, right, [1]H spectrum and [19]F spectrum, respectively). This demonstrated that changes in ligand binding, when T2M4SOD1 was present, could be followed by NMR spectroscopy.

While line broadening of the FUrd resonances was noticeable, there were relatively small chemical shift changes of the FUrd protons (>0.03 ppm, Table 4) leading us to conclude that

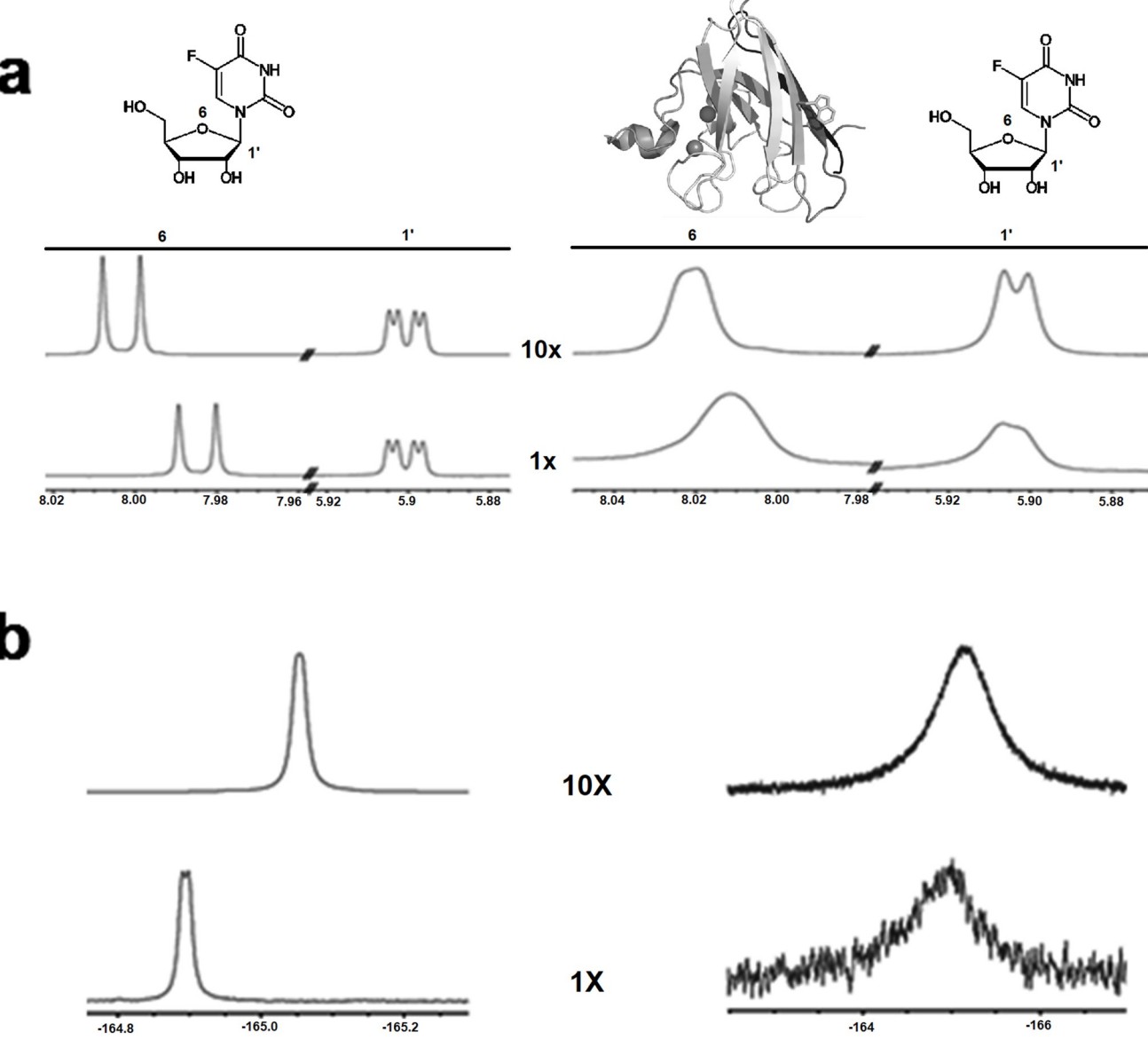

**Fig 4. ¹H-NMR and ¹⁹F-NMR of 5-fluorouridine (FUrd) alone and with T2M4SOD1.** (a) ¹H-NMR of FUrd (left) and FUrd in the presence of T2M4SOD1 (right). (b) ¹⁹F-NMR of FUrd (left) and FUrd in the presence of T2M4SOD1 (right). The spectra show the ligand and protein in a 1:1 ratio (protein: ligand-1X) and in a 1:10 ratio (protein: ligand-10X). The #1' is the ¹H of the ribose while the #6 is the ¹H of the fluorouracil moiety. Alone, the FUrd showed stacking as the there was a shift of ¹H #6 with increase in concentration (1X compared to 10X). The sugar moiety of the ligand was not influenced as there was no change in chemical shift of the ¹H #1'. When T2M4SOD1 was added to FUrd, there was line broadening (15.2 and 8.3 Hz for 1X and 10X, respectively, Table 2). (b) The ¹⁹F spectrum of FUrd alone (left) confirmed that it tended to stage or stack (Table 3, 14 Hz). In the presence of T2M4SOD1, line broadening was seen (667 Hz and 515 Hz, for 1X and 10X, respectively, Table 3) validating the interactions between protein and ligand. The 44-fold to 62-fold increased broadening of the ¹⁹F signal compared to ¹H #6' suggested a key-role of ¹⁹F in the binding process.

the binding was very weak and that the equilibrium of the binding was mainly on the side of the free ligand and free protein rather than bound ligand-protein. This agrees with the findings of the X-ray crystallographic studies, as FUrd could not be co-crystalized with the I113T SOD1 mutant. Indeed, the FUrd could only be seen bound to the protein after extensive soaking of the compound with the protein crystals. To strengthen the conclusions made with the ¹H data, we measured the ¹⁹F chemical shift and linewidth changes when T2M4SOD1 was present. We

**Table 2. Quantification of $^1$H line broadening of ligands (Hz) of the uracil ring proton #6, with and without T2M4SOD1.**

| Ligand | NMR Linewidth (Hz) | | | |
|---|---|---|---|---|
| | Without protein | Ligand: Protein ratio 1:1 | Ligand: Protein ratio 10:1 | Ligand: Protein ratio 50:1 |
| Urd | 0.8–0.9 | 0.8 | 0.9 | 0.8 |
| FUrd | 0.9–1.0 | 15.2 | 8.3 | ND |
| BrUrd | 0.8–1.1 | 1.4 | 1.2 | 1.2 |
| F3TDR | 1.4 | 3.2 | 3.1 | 2.8 |
| U | 0.7 | Too broad to measure (>100 Hz) | 26.5 | 29.5 |
| FU | 0.8–0.9 | Too broad to measure (>100 Hz) | Too broad to measure (>100 Hz) | Too broad to measure (>100 Hz) |
| BrU | 0.8–1.1 | Too broad to measure (>100 Hz) | Too broad to measure (>100 Hz) | Too broad to measure (>100 Hz) |
| ClU | 0.7–0.8 | Too broad to measure (>100 Hz) | Too broad to measure (>100 Hz) | Too broad to measure (>100 Hz) |

In case of a multiplet, a median was determined and used.

ND-not done.

found that the $^{19}$F signal from FUrd exhibited very significant line broadening (667 Hz and 515 Hz for 1X and 10X, ligand to protein, respectively; Table 3) but exhibited nearly unchanged chemical shifts (same as $^1$H chemical shifts; Table 5). The $^{19}$F data is in excellent agreement with the data derived by from our $^1$H experiments. These data suggest that the binding is localized to the $^{19}$F atom on the uracil moiety (Fig 4B; Tables 2 and 3). Because there were significant line broadening and only small chemical shift changes, we confirmed that FUrd was a relatively weak binder with an intermediate exchange rate between the bound and the unbound state.

Next, we tested if other halogenated derivatives of FUrd or Urd interacted with T2M4SOD1. Surprisingly, we found no evidence that Urd interacted with T2M4SOD1. There was no line broadening or chemical shift changes in Urd (1X, 10x or 50X) when T2M4SOD1 (1X) was present (Tables 2 and 4). Substantially less line broadening was seen with F3TDR (3.2 Hz, 3.1 Hz and 2.8 Hz with 1X, 10X and 50X ligand to 1X protein) compared to FUrd (15.2 Hz and 8.3 Hz with 1X and 10X ligand; Table 2). The third derivative, BrUrd, showed minimal binding as even less line broadening (1.2 Hz) was seen compared to FUrd. The $^{19}$F-NMR data of F3TDR showed markedly less line broadening (8.6 Hz and 6.8 Hz for 1X and 10X; Table 3) compared to the FUrd (667 Hz and 515 Hz for 1X and 10X ligand). Similarly, there were no chemical shift changes with Urd and very small chemical shifts changes with both BrUrd and F3TDR compared to FUrd (i.e., -0.025 ppm compared to -0.004 ppm and -0.01 ppm for BrUrd and F3TDR, respectively: Table 4). These data suggested that bromine, a larger halogen atom than fluorine, in BrUrd, and the three fluorines of F3TDR, sterically hindered and likely interfered with the binding to T2M4SOD1.

**Table 3. Quantification of $^{19}$F line broadening of the uracil aromatic ring proton #6 (substituted with $^{19}$F) with and without T2M4SOD1.**

| Ligand | NMR Linewidth (Hz) | | | |
|---|---|---|---|---|
| | Without protein | Ligand: Protein ratio 1:1 | Ligand: Protein ratio 10:1 | Ligand: Protein ratio 50:1 |
| FUrd | 14 | 667 | 515 | ND |
| F3TDR | 3 | 8.6 | 7.8 | 6.8 |
| FU | 10–15 | Too broad to measure | 300 | 280 |

In case of a multiplet, a median was determined and used.

ND-not done.

**Table 4. Quantification of the $^1$H differences in chemical shifts of the uracil aromatic ring proton #6 with T2M4SOD1.**

| Ligand | NMR Chemical Shift (ppm) | | |
|:---:|:---:|:---:|:---:|
| | Ligand: Protein ratio 1:1 | Ligand: Protein ratio 10:1 | Ligand: Protein ratio 50:1 |
| Urd | >0.001 | >0.001 | >0.001 |
| FUrd | -0.025 | -0.017 | ND |
| BrUrd | -0.004 | -0.005 | -0.003 |
| F3TDR | -0.01 | -0.009 | -0.006 |
| U | -0.004 | -0.007 | -0.008 |
| FU | Undetectable | Undetectable | Undetectable |
| BrU | Undetectable | Undetectable | Undetectable |
| ClU | Undetectable | Undetectable | Undetectable |

All values are in ppm. In case of a multiplet, the center resonance was calculated and used.

ND-not done.

These data strongly suggested that the uracil moiety of 5-FUrd was interacting with T2M4SOD1. Therefore, we investigated whether uracil alone or different halogenated versions of uracil bound T2M4SOD1. We found that uracil (without any halogenated substitutions) bound more strongly to T2M4SOD1 (Fig 5A) than FUrd as seen by the large increase in the $^1$H-linewidth (26.5 Hz for 10X Uracil compared to 8.3 Hz for FUrd; Table 2). By substituting the proton at position 5 with other halides (F, Cl and Br atoms), the remaining $^1$H signal of the uracil ligand broadened tremendously (beyond observation) suggesting increased binding by all halogenated versions of uracil (5-FU shown in Fig 5B, Tables 2–5). The $^{19}$F-NMR data of 5-FU showed significant line broadening (300 Hz for 10X ligand; Table 3) but not as broad as FUrd (515 Hz for 10X ligand; Table 3). These experiments strongly suggest that the uracil moiety of FUrd is interacting with T2M4SOD1.

To further explore these interactions, we examined if there was a transfer of magnetization between the protein and the different pyrimidine ligands (the uracil and uridine derivatives) using STD spectral difference. Because no transfer of magnetization was observed between ligands with T2M4SOD1, this confirmed the weak binding and intermediate exchange rate observed in the initial 1D $^1$H- and $^{19}$F - NMR spectra. These findings are particularly interesting when compared with the results with BrUrd and 5-BrU, suggesting that the 2-ribose moiety affects binding. The presence/absence of ribose leads to large differences in binding between the uracil derivatives and the uridine derivatives.

The above ligand studies were performed using T2M4SOD1 purified from cultures grown in rich LB-medium that was shown to be enzymatically active by an in-gel enzymatic assay. We performed parallel experiments with the T2M4SOD1 purified from cultures grown in M9 medium without metals and 1X, 10X and 50X 5-FU. The same $^1$H-NMR or $^{19}$F-NMR spectral

**Table 5. Quantification of $^{19}$F chemical shift changes of the uracil aromatic ring proton #6 (substituted with $^{19}$F) for different ligands added to T2M4SOD.**

| Ligand | NMR Chemical Shift (ppm) | | |
|:---:|:---:|:---:|:---:|
| | Ligand: Protein ratio 1:1 | Ligand: Protein ratio 10:1 | Ligand: Protein ratio 50:1 |
| FUrd | -0.025 | -0.017 | ND |
| F3TDR | -0.01 | -0.009 | -0.006 |
| FU | Undetectable | Undetectable | Undetectable |

All values are in ppm. In case of a multiplet, the center resonance was calculated and used.

ND-not done.

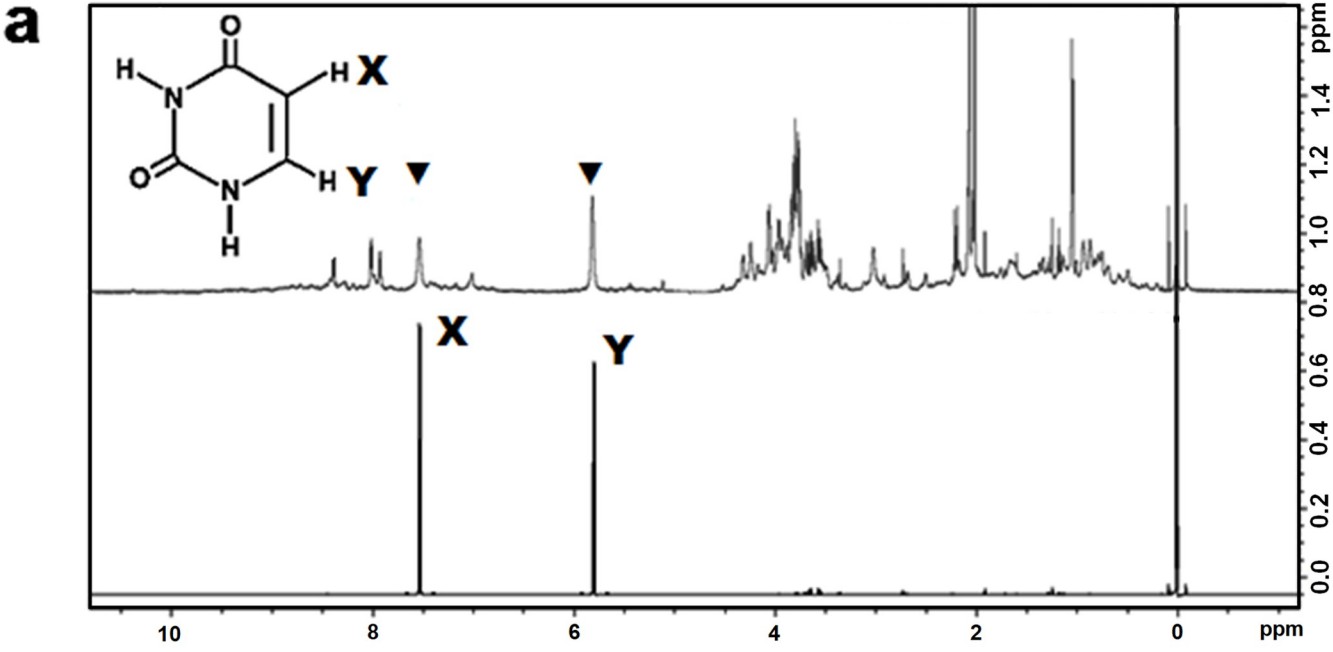

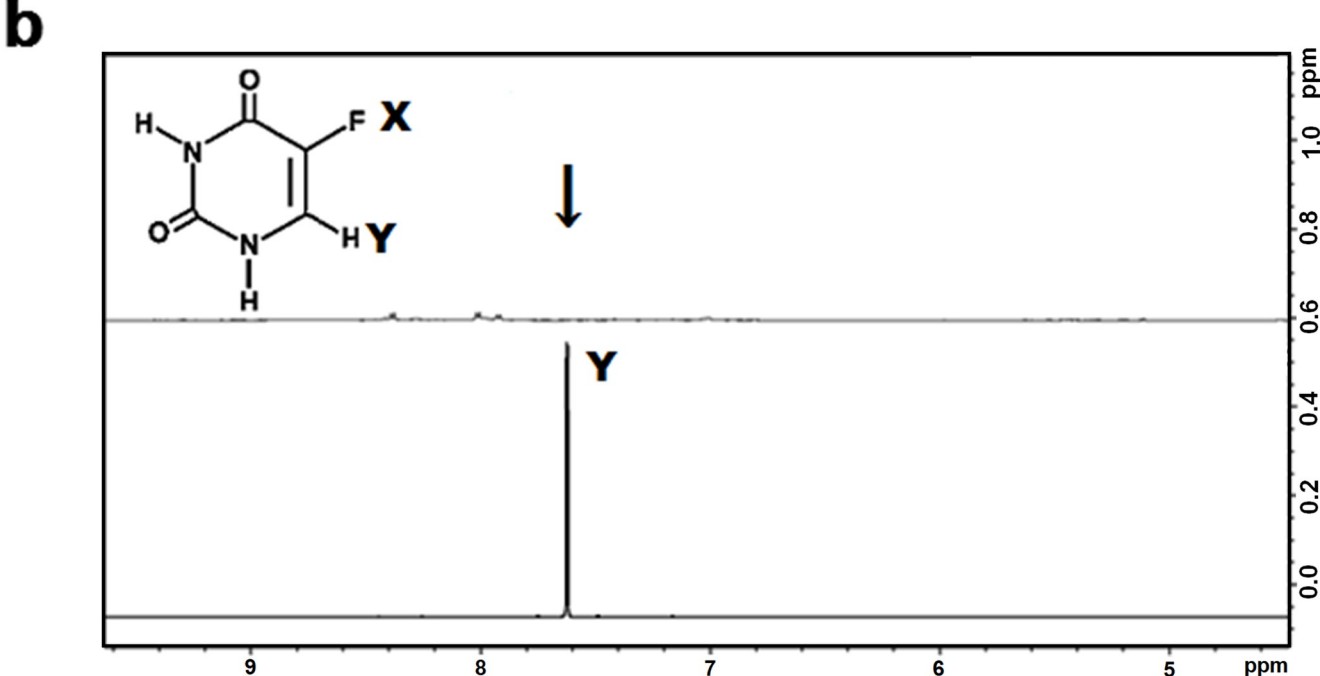

**Fig 5.** $^1$H-NMR of (a) uracil (U) or (b) fluorouracil (FU) alone and in the presence of T2M4SOD1 (each panel, bottom spectra- ligands alone; top spectra-ligand with T2M4SOD1). There was broadening (26-fold difference for U, ▾) or loss of signals (> 26 fold difference for 5-FU, ↓) when T2M4SOD1 was added. X, Y = H for uracil; X = F; Y = H for fluorouracil.

shifts and line broadening were seen with 5-FU as was observed with T2M4SOD1 purified from cultures grown in LB with metal salts. This suggests that the T2M4SOD1 purified from cultures grown without additional $Zn^{2+}$ and $Cu^{2+}$ can be used in the NMR-ligand studies. We

tested whether changes in Trp32 were detectable by NMR when [15]N-labelled T2M4SOD1, purified from M9 medium without metals, was mixed with 10-fold excess FUrd, 5-FU (as positive controls) or Urd (as a negative control). By NMR, no changes in any residue were observed with any ligand when compared to [15]N-labelled T2M4SOD1 alone (Fig 6). Using a published [1]H-[15]N-HSQC of Banci et al., 2006 [39] as a reference (Biological Magnetic Resonance Data Bank- BMRD entry 6821), we identified a few amino acids of the [15]N-labelled T2M4SOD1 (Fig 6). The indole proton of Trp 32 (bottom left) was easily identified while the Asp 96 residue was not readily identified as it is clustered around several peaks. No changes were detected in the indole protons of Trp32. These results further substantiate that FUrd/ F-FU binding is weak and that the equilibrium with T2M4SOD1 favors the free rather than the bound ligand.

## Molecular docking and simulation ligand studies

Since our NMR-ligand experiments could not directly confirm the binding of the uracil derivatives to Trp32, we used molecular docking, molecular dynamics simulations and DFT chemical shift calculations to help us identify putative binding sites on SOD1. Our molecular docking and MD simulation experiments used the well-studied SOD1 mutant I113T complexed to FUrd and our computational results generally agreed with our NMR experimental findings (with FUrd, FU, ClU, and BrU predicted to bind most strongly). In particular, using docking simulations, the lowest negative energy was detected with FUrd, FU, ClU and BrU when the largest cluster criteria were employed (Table 6). More positive binding energies were detected with the other uridine derivatives and with FT3DR, suggesting weaker binding interactions. However, it is important to note that these binding energies are not vastly different from each other. Furthermore, several binding sites for all the ligands over the SOD1 surface were identified during our docking runs, suggesting that there may not be a single preferred binding site for these ligands.

We then used MD simulations to identify binding interactions with I113T SOD1. During the production (long-term) MD simulation, the distance between the center of mass (COM) of the ligand and the COM of the Trp32 side chain varied considerably (Fig 7). For most of the simulation time, all ligands had no π- π interactions with the Trp32 side chain as no oscillations between 0.25–0.75 nm were detected (Fig 7A–7H). However, BrU, ClU, FU and U (Fig 7A–7D) interacted occasionally (6, 6, 1 and 1 time, respectively) with Trp32 as evidenced by oscillations that approached 0.25–0.75 nm during the 100,000 ps time frame. This suggested that BrU, ClU, FU and U may form π-π stacking conformations with Trp32. Because the Trp32 interactions were not observed with the uridine derivatives, which contain the ribose moiety, this suggested that the ribose moiety may be binding to other amino acids within SOD1. The MD simulations also suggested that there was no specific binding site in the I113T SOD1 structure for these ribosylated ligands since the 0.25–0.75 nm COM contacts were absent or very infrequent.

We then employed DFT with the M06-2X density functional to identify putative binding interactions with amino acids around Trp32. To properly model this interaction at the quantum mechanical level, docked conformations in the closest proximity of Trp32 were optimized using the density functional M06-2X/6-31G**. Ligands without a ribose moiety (uracil derivatives, Fig 8A–8D) showed prominent interactions with the aspartate (Asp96) side chain of SOD1 and the H[N/C] of the pyrimidine ring. These ligands were held in the binding site via H-bonding interaction with Asp96 side chain. For the ribose-containing ligands (uridine derivatives, Fig 8E), such interactions shift to the Asp96 side chain and uracil hydroxyl group. These calculations prove that H-bonding interactions help direct the

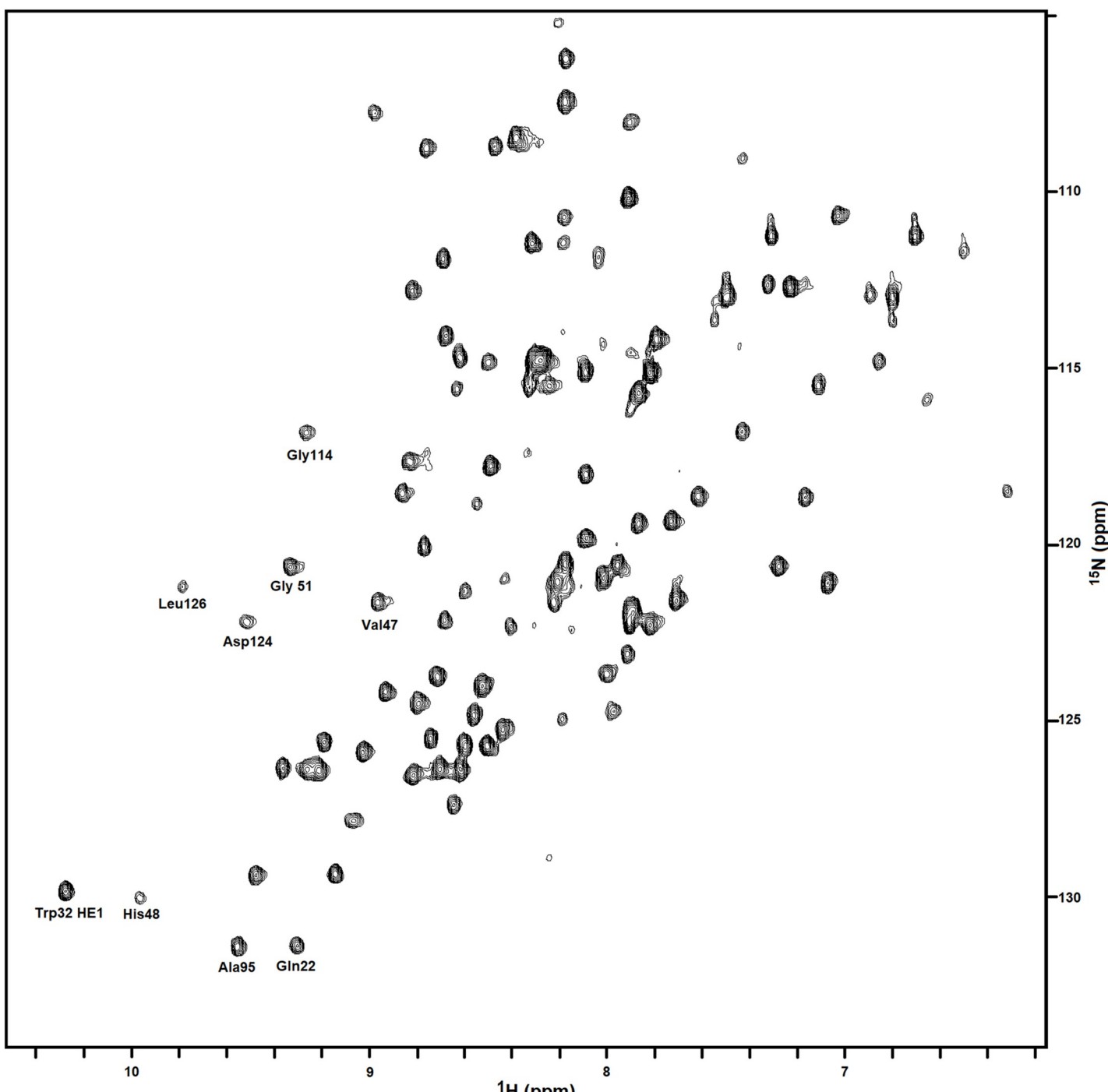

**Fig 6. ¹H -¹⁵N Heteronuclear Single Quantum Coherence (HSQC) of the T2M4 SOD monomer only.** The lyophilized T2M4 monomer, purified from cultures grown with ¹⁵NH₄Cl, was dissolved to 440 μM in 20 mM potassium phosphate buffer, pH 7.4 containing 10% D₂O. When compared to the ¹H-¹⁵N-HSQC of the monomer alone, there was no changes in the chemical shifts of any amino acid residues when pyrimidines were added. Some amino acid residues were identified comparing to the ¹H-¹⁵N-HSQC data uploaded by Banci et al. [39] into the Biological Magnetic Resonance Data Bank entry 6821.

ligand binding with SOD1 and suggest that Asp96 may be more likely involved in binding the uracil ligands though the stronger H-bonding rather than the weaker π- π interactions with the Trp32 side chain.

**Table 6. Calculated docking energies (in kcal/mol) of different ligands bound to SOD1 (PDB ID: 4A7S) compared to the measured $^1$H NMR line broadening of the #6 proton of uracil.**

| Ligand | Docking (kcal/mol, largest cluster, 2.5 Å) | $^1$H NMR Line Broadening (Hz) |
|---|---|---|
| BrU | -3.71 | >100 |
| ClU | -3.58 | >100 |
| U | -3.29 | >100 |
| FU | -3.28 | >100 |
| FUrd | -3.25 | 15.2 |
| BrUrd | -3.20 | 1.4 |
| Urd | -3.20 | 0.8 |
| F3TDR | -2.48 | 3.2 |

We also examined different complexes formed between 3-methylindole and BrU, using DFT chemical shift calculations. This was done to calculate expected chemical shift changes arising via possible π-stacking and hydrogen bond interactions at the quantum mechanical level (as full protein DFT chemical shift calculations are impractical). The 3-methylindole unit was used as a model for the Trp32 side chain and BrU was used as an example of the 3 halogenated uracils that exhibited extensive line broadening in our NMR experiments. Four different putative complexes were tested involving H-bonding, π-stacking or two different partial π-stacking interactions (S1 Fig top). A 1–1.5 ppm shift in the $^1$H isotropic chemical shielding tensors of hydrogen from pyridine ring was predicted in the π-stacked complexes formed between the indole side chain of Trp32 and 5-bromouracil molecule (S1 Fig bottom). Marginal chemical shift changes were predicted with the H-bonding complex formed between the indole side chain of Trp32 and 5-bromouracil molecule. This suggested that some π-stacking with Trp32 of SOD1 occurs with BrU binding (and likely the other halogenated uracils). These DFT-predicted chemical shift changes of about 1 ppm would explain the relatively large line broadening seen for the uracil #6 proton for these halogenated uracil derivatives.

## Discussion and conclusion

We successfully designed, cloned and purified a recombinant, thermostable, monomeric SOD1 protein (T2M4SOD1) from the periplasm of *E. coli* cultures BL-21 (DE3). This was done with and without added copper and zinc, two metals that are required for SOD1 activity and structure, respectively. While the M4SOD1 had been described and purified previously [13], many key details regarding the purification protocol were left out. Furthermore, the preparation of monomeric SOD1 has not been reproduced by any other group outside of the original authors. Furthermore, the yields were low, and the purification protocol required considerable amounts of cell culture (10 L). These problems led us to develop a better, more clearly defined expression system and an improved protein purification protocol with the intent of increasing yields and permitting more widespread use of this important model protein in structural studies relating to ALS and ALS drug discovery.

As shown from our results, we found that the use of an *E. coli* codon optimized DNA sequence, the incorporation of a better leader sequence for enhanced protein export (the leader peptide from bacteriocuprein), a stronger promoter (T7 instead of tac), a better bacterial host (BL21-DE3), more optimal growth conditions (better media recipes and incubation temperatures) and more controlled cell lysis, greatly improved protein yields (by a factor of 10-20X). We have elaborated on these details in an effort to provide the community with a clear and reproducible protocol for the high-yield production and purification of monomeric SOD1 in *E. coli* from both rich and minimal media. Through this work we determined that the

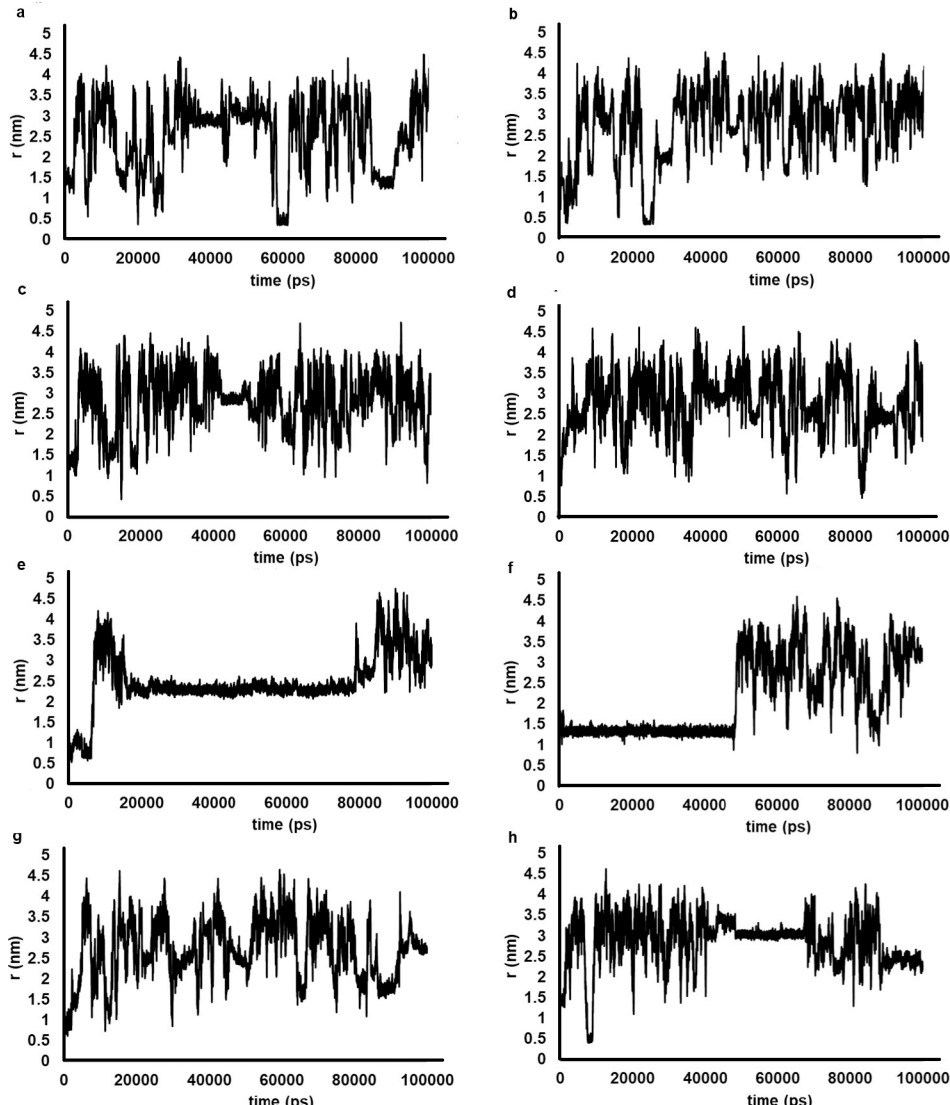

**Fig 7. Distance scan from the center of mass of the Trp32 side chain and the uracil ligands for last 100 ns of production simulation.** The BrU (a), ClU (b), FU(c) and U (d) ligands showed possible π-π interactions with the Trp32 aromatic side chains. During most of the production simulations. uridine ligands (e-h, BrUrd, FT3Dr, FUrd, and Urd, respectively), containing ribose moieties, had no noticeable π-π interaction between the aromatic side chains of the Trp32 residue.

T2M4SOD1 could be abundantly expressed after overnight expression at RT and 28˚C but not at 37˚C and only in rich medium when $Zn^{2+}$ and $Cu^{2+}$ were added. We found that T2M4SOD1 was very poorly expressed in minimal medium when $Zn^{2+}$ and $Cu^{2+}$ metal salts were explicitly added to induced cultures. However, the T2M4SOD1 protein could be abundantly expressed, purified and was active (and presumably properly metallated), when prepared from minimal M9 medium without the addition of metal salts. Therefore, isotopically labelling of T2M4SOD1 in minimal medium can only be achieved without the addition of metal salts.

It is questionable whether $Cu^{2+}$ and/or $Zn^{2+}$ must be added to *E. coli* cultures to ensure that the recombinant monomeric SOD1 is properly metallated. Banci et al. [13] purified the

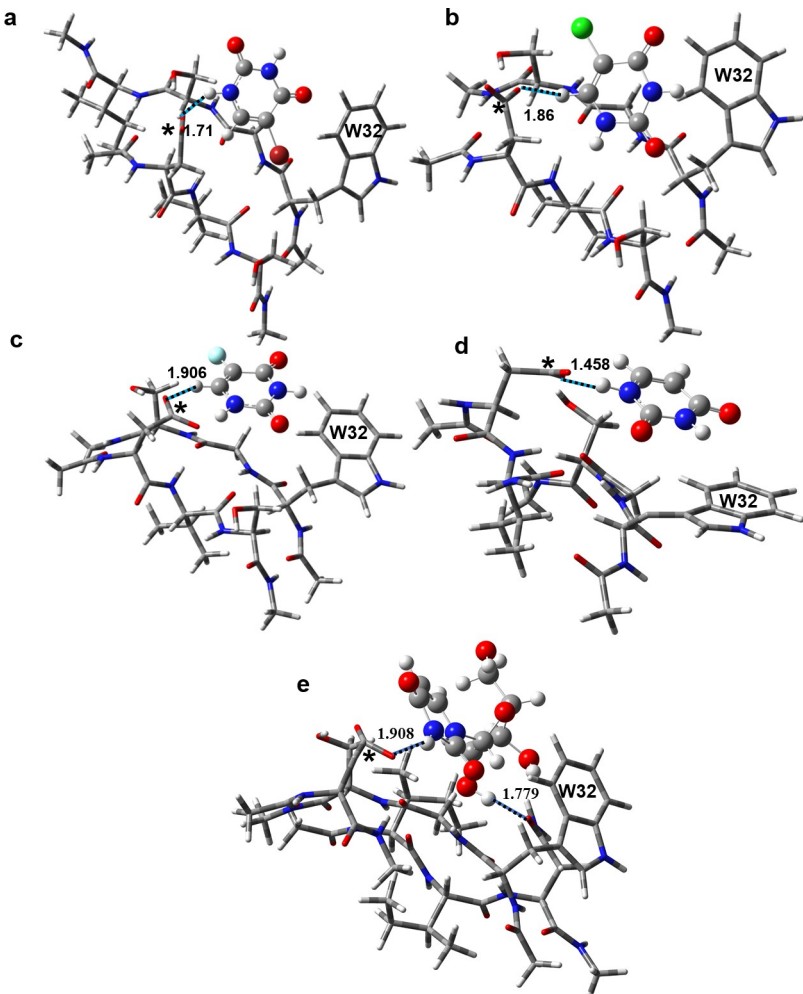

**Fig 8.** Optimized geometries of the uracil ligands and uridine at the binding site close to the Trp32 side chain computed at the M06-2X/6-31G** level of theory: (a) BrU, (b) ClU, (c) FU, (d) U and (e) Urd. The ligands are shown in the ball and stick model, and the receptor residues are in the tube model. The aspartate 96 residue, making H-bonding contacts with ligands (dashed lines), is marked with an *. The H-bond distances are in Å. W32 indicates the tryptophan 32 residue. Atom colour code: H: White; C: Grey; Nitrogen: Blue; F: Aqua green; O: Red; Cl: Green; Br: Magenta.

T2M4SOD1 from cultures grown in 10 L of L-broth with 100 μM copper sulfate. Tachu et al. [40] purified recombinant wild-type dimeric SOD1 and fALS variants using a PelB leader peptide fused to SOD1. These recombinant SOD1 proteins from Tachu et al. were exported to the periplasm of E. *coli* BL-21 (DE3) from cells grown in LB medium with 166 μM each of copper chloride and zinc chloride. Lin et al. [41] examined the expression and activity of a human dimeric SOD1 fused to a histidine tag in E. *coli* BL-21 (DE3) cultures grown in LB medium with 30–1000 μM $Cu^{2+}$ and 10–100 μM $Zn^{2+}$. The expression of their recombinant dimeric SOD1 was unaffected by increasing concentrations of $Cu^{2+}$ or $Zn^{2+}$ while their recombinant dimeric SOD1 activity in crude bacterial lysates, measured by the absence of reduction of their tetrazolium salt was increased with 750 μM $Cu^{2+}$ and 15–30 μM $Zn^{2+}$. These results are consistent with our observations regarding the expression of monomeric SOD1 in E. *coli* grown on LB media. However, high concentrations of $Zn^{2+}$ and $Cu^{2+}$ are toxic to E. *coli* as we observed with E. *coli* cultures grown on minimal medium. E. *coli* uses transporters to control $Zn^{2+}$ and

$Cu^{2+}$ concentrations within the cell as these heavy metals are important cofactors for bacterial metalloproteins and metalloenzymes but can be toxic at high concentrations. When excess $Zn^{2+}$ is present, the expression of $Zn^{2+}$ transporters is repressed [42]. Chandrangsu et al. [43] reviewed metal homeostasis in bacteria and reported that the intracellular pool of zinc could be 1 mM. Most of the cellular copper in *E. coli* in located in the periplasm as $Cu^{2+}$ and *E. coli* actively exports any $Cu^{2+}$ from the cytoplasm to the periplasm [42]. The periplasmic $Cu^{2+}$ concentration is sufficient for $Cu^{2+}$ storage when low $Cu^{2+}$ concentrations (0.1–50 μM) are in the culture medium [44] and may be sufficient to transfer the $Cu^{2+}$ sequestered there to our T2M4SOD1. Our data suggests that the T2M4SOD1 was correctly metallated in rich media as well as minimal media. Addition of metals to growing *E. coli* BL-21 (DE3) cultures may be unnecessary since there appears to be sufficient $Cu^{2+}$ or $Zn^{2+}$ concentrations intracellularly. Moreover, Kirsten et al. [45] quantified metals in LB medium using ICP/MS, and found that sufficient $Zn^{2+}$ and $Cu^{2+}$ are present in LB alone to support growing *E. coli* cultures at optical densities measured at 600 nm ($OD_{600}$) equal to 3 or 7 for $Cu^{2+}$ and $Zn^{2+}$, respectively.

After expression and initial isolation, both the leader-peptide-containing lpT2M4SOD1 and leader-peptide-free T2M4SOD1 were observed by Coomassie-staining of Tricine SDS-PAGE gels of whole cells, suggesting that all the expressed protein did not get sorted to the periplasm. Therefore, it is important that periplasmic proteins be isolated from fresh-not-frozen cells as the contents of the cytoplasm, where the leader-peptide containing T2M4SOD1 resides, could be released upon a freeze-thaw cycle. Interestingly, Tachu et al. [40] expressed and purified recombinant wild-type and mutant fALS SOD1 variants from *E. coli*, each fused with the PelB signal sequence, which sorted the dimeric SOD1 to the periplasm. The authors purified their dimeric SOD1 proteins from the periplasm of frozen cells. They effectively reduced the enrichment that strict periplasmic purification affords as cytoplasmic proteins are in 10-fold excess of periplasmic proteins [46]. We also note that the Coomassie stained periplasmic extracts published by Tachu et al. show a much higher amount of the leader peptide containing recombinant SOD1. We purified the T2M4SOD1 from freshly pelleted cells to prevent any cytoplasmic proteins from entering the periplasmic pool.

There are several known methods to extract periplasmic proteins from *E. coli*. Osmotic shock, lysozyme-EDTA, polymyxin digestion and chloroform extraction have been shown to recover periplasmic target proteins well, but these methods also released cytoplasmic proteins [46]. Selecting a method to release the contents of the periplasm without contamination of the cytoplasmic proteins is an important first step in the purification of recombinant proteins from the periplasm. As noted before, previously published methods were ambiguous in their descriptions [13,15] regarding osmotic shock methods. Quan et al. [46] using a Tris-sucrose-EDTA buffer for periplasmic extraction, observed that different periplasmic proteins were released differently in various *E. coli* strains suggesting that one method may be more efficient in one *E. coli* strain than another. Banci et al. [13] as well as Getzoff et al. [15] both expressed recombinant SOD1 in *E. coli* strain MC1061, while we expressed and purified our T2M4SOD1 from *E.coli* BL-21 (DE3). The BL-21 strain is deficient in proteases that are known to degrade foreign proteins [47]. Our combined method of extracting T2M4SOD1 using sucrose:EDTA followed by osmotic shock with $CuSO_4$ released T2M4SOD1 without releasing the leader-peptide containing SOD1.

We further purified the T2M4SOD1 by heating the pooled sucrose/EDTA and osmotic shock fractions followed by $(NH_4)_2SO_4$ precipitation and anion exchange chromatography. This protocol was inferred from partial descriptions provided by Beyer et al. [12] and Koshland and Botstein [20]. Heating the protein extract at 65°C will precipitate some thermolabile proteins that can be removed by centrifugation. Several proteins were pelleted after heating the T2M4SOD1-containing proteins extracted from minimal medium, including the T2M4SOD1.

The T2M4SOD1 was not pelleted after heating extracts from LB-grown cells, suggesting that the SOD1 is more labile from minimal medium grown cells. When removing non-SOD1 proteins using $(NH_4)_2SO_4$, the 50% $(NH_4)_2SO_4$ cut successfully removed other proteins without precipitating the T2M4SOD1. However, with 65% $(NH_4)_2SO_4$ and 100% $(NH_4)_2SO_4$, the T2M4SOD1 was also precipitated, suggesting that as little as 50% $(NH_4)_2SO_4$ can be used to partially purify the T2M4SOD1. This contrasts with the referenced protocol that purified the recombinant human SOD1 using the supernatant of the 100% $(NH_4)_2SO_4$ precipitation. Therefore, the T2M4SOD1 was easily purified by periplasmic extraction using sucrose/EDTA, then osmotic shock, heating, pelleting, 50% ammonium sulfate precipitation followed by anion-exchange chromatography.

The NMR titration studies of SOD1 with uracil derivatives supported the binding of uracil (alone) and halogenated uridine compounds to T2M4SOD1. Because no changes in the NMR spectra of uridine were seen when mixed with SOD1, this indicated that uridine was not able to bind to the protein. While X-ray data suggests the binding occurs at Trp32, we did not observe changes in Trp32 when the [15]N-labelled T2M4SOD1 was mixed with FUrd, 5-FU, the positive controls or uridine, the negative control. As noted before, X-ray crystallography studies revealed that uridine-5′-monophosphate (UMP) binds at the groove between the electrostatic and the Zn-binding loops of SOD1 [48], whereas FUrd bound at the Trp32 site of SOD1 [9]. The current NMR titration studies suggest that halogenation of 5th position of the uracil ring is important for the binding of uracil nucleosides to T2M4SOD1. Overall, our NMR titration studies support that nucleoside and mononucleotide forms of uracil weakly bind with T2M4SOD1 but that the binding is not occurring exclusively or tightly at the Trp32 site.

The first observations of interaction of uracil derivatives with SOD1 were described by Antonyuk et al. [48]. They soaked crystals of dimeric L38V SOD1 protein with UMP and saw interactions near Trp32 and the Zn-binding loop. Similarly, when I113T mutant crystals were soaked with FUrd [9], binding was also observed near Trp32. It should be noted that co-crystallization of ligands with mutant SOD1 proteins was not possible. More recently, a low affinity micromolar binding to Trp32 was demonstrated with a naphthalene-catechol-linked compound complexed with crystals structures of SOD1 [49]. This suggests that the binding of these ligands with Trp32 is not tight.

Molecular docking and molecular dynamics studies helped us determine how Trp32 and/or other amino acids in SOD1 interacted with the uracil and uridine-based ligands. The docking simulations correctly predicted the strength of the binding interactions as measured by the linewidths observed by NMR (Table 6). However, the binding energies were not found to be particularly strong. Furthermore, the calculations suggested several binding sites over the SOD1 receptor surface with comparable docking energies for all the ligands. These multiple binding sites indicate that there may not be a single preferred binding site over other sites, for these ligands. This could explain the absence of a detectable change in the Trp32 resonance in the [15]N-NMR spectra. While weak π-π stacking with the 5-membered pyrrole ring, but not the benzene ring of Trp32, was observed with bromouracil, H-bonding via Asp 96 was the more dominant interaction for all uracil ligands. However, the observed π-π stacking interaction could result from multiple H-bonding interactions involving different sites of SOD1 and the ligands. In particular, it was recently demonstrated that water molecules located between Glu21 and Lys30, and between Asp96 and Ser98 residues mediate H-bonding between 5-FUrd and the SOD1 protein [10]. Such interactions may potentially position the 5-fluorouracil group of 5-FUrd on the top of the indole group of Trp32. Our MD simulations of complexes with unrestricted ligands provide evidence of the very weak nature of the π-stacking in the present context. Indeed π-stacking interactions were absent for most of the ligands during our MD runs. The importance of binding-site water molecules for docking simulations in

reproducing the receptor bound-conformation of the ligand was reported previously [10]. However, unlike a single, static snapshot provided by a crystal structure, the dynamics of protein-ligand interactions are better represented by liquid phase simulations that describe time evolution of different conformational ensembles (and hence position of ligands around a receptor).

The nature of the interactions of uridines and uracils with Trp32 could be due to the involvement of other amino acids or other structural features. Our calculations show that the uracil ligands, which are pyrimidines, are smaller in size and are held in the SOD1 binding site by forming a strong H-bond with the Asp96 side chain. The larger uridine ligands, with an added ribose ring, also form H-bonds with the Asp96 side chain and the pyrimidine ring. They also form additional H-bonds with the ribose hydroxyl groups and protein backbone amide carbonyl groups as well as other nearby amino acid side chains. Because conventional H-bonding interactions are much stronger than $\pi$-stacking interactions, the most stable ligand-protein complex would be predicted to involve hydrogen-bonding interactions when both possibilities are present. These additional H-bonding interactions place the pyrimidine ring away from the Trp32 benzene ring and this likely explains the absence of NMR chemical shift changes in the uracil chemical shifts arising from $\pi$-stacking interactions (as seen in our NMR experiments).

Because wtSOD1 is naturally a dimeric protein, it makes ligand studies of this protein by NMR spectroscopy difficult. This is because duplicate NMR signals must be painstakingly sorted out to infer clear structural changes. By developing a robust, thermostable, monomeric SOD1 construct (T2M4SOD1), a substantially improved route for expressing and purifying this protein (in both rich and minimal media) and showing how such a construct can be studied by NMR, we believe we have helped open the door to other labs to perform NMR-based structural studies with monomeric SOD1. Furthermore, such an SOD1 construct could be particularly amenable to the production of fALS variants and for the testing of various ligands or drug leads that could help prevent the misfolding and propagation of SOD1 in ALS.

## Supporting information

**S1 Fig. Different potential complexes formed between the indole and 5-Bromo uracil (upper) and GIAO/M06-2X/6-311++G$^{**}$ calculated NMR shielding tensors of the pyrimidine ring hydrogen in complexes between the free 5-Bromo uracil and in complex with the indole derivative.**
(TIF)

**S1 Raw images.**
(PDF)

## Acknowledgments

We thank Dr. Mark Bejanskii for useful advice and discussions. We are grateful for assistance from Dr. Randy Whittal (Manager) and Béla Reiz (Technologist) of the Mass Spectrometry Facility, Department of Chemistry, University of Alberta. Elise Beaton is thanked for graphic support.

## Author Contributions

**Conceptualization:** Marcia LeVatte, Matthias Lipfert, Andriy Kovalenko, David Scott Wishart.

**Data curation:** Matthias Lipfert.

**Formal analysis:** Dipankar Roy.

**Funding acquisition:** Andriy Kovalenko, David Scott Wishart.

**Methodology:** Marcia LeVatte, Matthias Lipfert, Dipankar Roy.

**Supervision:** Andriy Kovalenko, David Scott Wishart.

**Visualization:** Marcia LeVatte, Matthias Lipfert.

**Writing – original draft:** Marcia LeVatte, Matthias Lipfert, Dipankar Roy, David Scott Wishart.

**Writing – review & editing:** Marcia LeVatte, Matthias Lipfert, Dipankar Roy, Andriy Kovalenko, David Scott Wishart.

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
