## [Decision Letter · Decision Letter 0]

19 Jan 2021

PONE-D-20-39700

Cloning and high-level expression of monomeric human superoxide dismutase 1 (SOD1) and its interaction with pyrimidine analogs

PLOS ONE

Dear Dr. Wishart,

Thank you for submitting your manuscript to PLOS ONE. After careful consideration, we feel that it has merit but does not fully meet PLOS ONE’s publication criteria as it currently stands. Therefore, we invite you to submit a revised version of the manuscript that addresses the points raised during the review process.

The manuscript was examined by two reviewers and both of them found the work interesting and technically sound. Yet, I kindly ask you to address the minor issues raised by the reviewers before we can proceed further with the publication.

We look forward to receiving your revised manuscript.

Kind regards,

Oscar Millet

Academic Editor

PLOS ONE

Journal Requirements:

"This  work was supported by funding from the Alberta Innovates Alberta Prion Research Institute (Research team program ABIBS APRIRTP 201300023 and explorations program ABIBS APRIEP 201600034; https://albertainnovates.ca/programs/alberta-prion-research-institute/). DR and AK acknowledge generous computing time provided by WestGrid (www.westgrid.ca) and Compute Canada/Calcul Canada (www.computecanada.ca).The funders had no role in study design, data collection and analysis, decision to publish, or preparation of the manuscript."

We note that you received funding from a commercial source: Compute Canada/Calcul Canada.

Reviewers' comments:

Reviewer's Responses to Questions

**Comments to the Author**

1. Is the manuscript technically sound, and do the data support the conclusions?

Reviewer #1: Yes

Reviewer #2: Yes

2. Has the statistical analysis been performed appropriately and rigorously? 

Reviewer #1: N/A

Reviewer #2: Yes

3. Have the authors made all data underlying the findings in their manuscript fully available?

Reviewer #1: Yes

Reviewer #2: Yes

4. Is the manuscript presented in an intelligible fashion and written in standard English?

Reviewer #1: Yes

Reviewer #2: Yes

5. Review Comments to the Author

Reviewer #1: Dear Authors,

Thanks for the opportunity to review the manuscript. I thoroughly evaluating this manuscript. I do have two minor suggestions that I would like you to address. First, with regards to the mutations that you engineered into SOD1 to make it more thermostable and exist in the monomeric state, are any of these mutations linked to ALS? In other words, can one consider your engineered SOD1 a 'wild-type' or 'mutant' enzyme with regards to disease? Second, the gel images would be much easier to interpret if the lanes were labeled with their contents, not just numbers.

Reviewer #2: This paper should be published without revision. The paper is well-written, well thought out, and its subject is important. The new monomeric form of SOD1 might prove to be useful to certain groups working on SOD1. While I point out a few issues that the authors could address, these are optional.

The paper creates a new monomeric model of SOD1 that will allow easier study with NMR, in particular, the study of protein-ligand interactions. This monomeric form is more stable (offers better yields in recombinant expression systems). These types of SOD1 models, or “Franken-SOD1’s”, as I like to call them, are oligo-mutated forms of SOD1. This one has 6 mutations to make it stable and monomeric, unlike the previous Franken-SOD1’s that had two or four mutations). This one is also not acetylated at the N-terminus (due to the expression in e. coli), so we could add a “seventh” chemical perturbation that moves it further and further away from naturally occurring SOD1 (in addition to the eighth perturbation, an ALS mutation). Here, I must be frank and honest with the authors. When one considers that the subtle switch of an R-OH group on the surface of SOD1, to an R-NH2 group (that is, the D101N mutation), results in one of the most severe forms of ALS, it should remind us to perturb the SOD1 protein as little as possible when studying it. This paper throws that prudent guideline of science completely out the window and speeds away! Their boldness is impressive, but I hope these are the last mutations we'll make for optimization. If you ever need even better yields, I would just scale up, use more flasks, and maybe buy another shaker, instead of adding more mutations. But maybe I am too conservative. I will also say that someone needs to contact David Borchelt at UF and try to convince him to create a transgenic mouse model for some of these Franken-SOD1’s. We need to make sure that they are actually “WT” and do not trigger ALS. For years, we have been working on these proteins, calling them “WT”, “pseudo-WT”. We should probably refer to them as NPMWH variants (Non-Pathogenic Mutants, We Hope)”

Issues to consider (optional):

1. Metals! What is the metal content? Is it properly metallated or does it mismetallate? ICP-MS and UV-Vis could help you get the answer quickly.

2. Intro: The authors state that there are conflicts between the works in references 8 and 9 (Lansbury and Hasnain) on whether certain molecules bind SOD1 or not, and that NMR can be an arbiter. They are right, NMR is the right/best tool to get a clear answer. But is there really a conflict between refs 8 and 9? One is an in silico prediction that does not attempt to model explicit solvent (ignoring entropic contributions), also performed at low (physiological) ionic strength, and the other is a crystallographic/experimental study, performed in molar quantities of salt (ammonium sulfate I presume), which increases the hydrophobic (solvent-entropy) effects and hydrophobic interactions that might not happen in physiological settings. Nobody can predict which, whether, and how molecules bind to proteins, de novo. This is why “drug design” is actually high-throughput screening.

3. Intro: No references on WT enhancing the toxicity of mutant SOD1 through heterodimerization (mutant enhancing WT is mentioned, but several mouse models suggest it might be the other way around!). The rate of mutant/WT sod1 heterodimerization is on the same timescale of its lifetime in vivo, and free energies have been measured for some mutants and metalation states.

6. PLOS authors have the option to publish the peer review history of their article (what does this mean?). If published, this will include your full peer review and any attached files.

Reviewer #1: No

Reviewer #2: No

---

## [Author Response · Author response to Decision Letter 0]

1 Feb 2021

February 1, 2021

Dear Dr. Millet,

Please find attached our revised manuscript (PONE-D-20-39700) entitled: ‘Cloning and high-level expression of monomeric human superoxide dismutase 1 (SOD1) and its interaction with pyrimidine analogs’ by Marcia LeVatte, Matthias Lipfert, Dipankar Roy, Andriy Kovalenko and myself. We want to thank the reviewers and editors for their comments and suggestions. We have made all the requested changes to Figure 2 and uploaded the changed file. We also deleted both of the ‘data not shown’ comments written in our original manuscript as these results were not a core part of the research being presented. These changes are reflected in the text. We have tried our best to address all the reviewers’ comments (bold, italicized text) and have provided details of our few modifications to the manuscript in the attached pages. We have also produced both a clean version and a marked-up version of the manuscript with all changes indicated in red so that the reviewers can better see the edits or modifications.

To the Editor’s comments: Journal Requirements:

Response: We checked our file names for figures, supplemental figures, raw data, manuscript, and rebuttal letter. We noticed that the figure files were not named correctly and removed a space or period between the g and number (from Fig 1.tif to Fig1.tif) to conform to the file naming style requirements for PLOS One. The raw data file was changed and now is correctly named (‘S1_raw_images’). 

The following 3 documents were uploaded with this revision following the naming convention required by the editor.

i. The rebuttal letter entitled 'Response to Reviewers'.

ii. A marked-up copy of the original manuscript with changes highlighted in red labeled 'Revised Manuscript with Track Changes'.

iii. An unmarked version of our revised paper without tracked changes labeled 'Manuscript’.

"This work was supported by funding from the Alberta Innovates Alberta Prion Research Institute (Research team program ABIBS APRIRTP 201300023 and explorations program ABIBS APRIEP 201600034; https://albertainnovates.ca/programs/alberta-prion-research-institute/). DR and AK acknowledge generous computing time provided by WestGrid (www.westgrid.ca) and Compute Canada/Calcul Canada (www.computecanada.ca).The funders had no role in study design, data collection and analysis, decision to publish, or preparation of the manuscript."

We note that you received funding from a commercial source: Compute Canada/Calcul Canada.

Response: Compute Canada/Calcul Canada appears to have been erroneously flagged as a commercial funding source. Both WestGrid and Compute Canada/Calcul Canada are Canadian government-funded organizations that provide Canadian university faculty members and students access to high performance computing and distributed data storage. The PLoS editorial staff may not have been familiar with the origins of these two Canadian computing resources. Since these are Canadian government funding/funded agencies, we did not amend our Competing Interests Statement because there were no competing interests to state.

Response: We did not add this statement as there were no competing interests to state (see above, all funding was from the government of Canada) and therefore there was no change in our ability to submit our article to PLOS One.

Please include your amended Competing Interests Statement within your cover letter. 

We will change the online submission form on your behalf.

Response: These changes do not need to be made (see above).

Response: As requested, the pdf file containing all the raw images for our gels are now in the Supporting Information, labeled as: ‘S1_raw_images’. This pdf was uploaded with all the manuscript files at the time of our re-submission.

Response: We have now removed all the ‘data not shown‘ instances in the manuscript. We mentioned ‘data not shown’ twice in our original manuscript. The ‘data not shown’ were found on lines 674 and 680 (page 33). We removed the ‘data not shown’ from line 674 as this was a confirmation that the SOD monomer, purified from cultures grown without metals, caused the same changes in the 1H- and 19F-NMR spectra of fluorouracil that was seen when the SOD monomer purified from cultures grown with metals. The results were the same and so this did not contribute to the core information in the manuscript. The ‘data not shown’ from line 680 was also removed because the 1H-15N-HSQCs generated from the monomer alone or monomer with each ligand (positive controls-fluorouridine or fluorouracil or negative control-uridine) were identical and unchanged. Like the first instance of ‘data not shown’, our belief is that these mentions of ‘data not shown’ do not contribute to the core information in the manuscript and so no additional data has been added to the document. We also note that none of the reviewers requested that the data be shown.

For the Reviewers' general comments:

Reviewer's Responses to Questions

Comments to the Author

1. Is the manuscript technically sound, and do the data support the conclusions?

Reviewer #1: Yes

Reviewer #2: Yes

Response: Thank you.

2. Has the statistical analysis been performed appropriately and rigorously?

Reviewer #1: N/A

Reviewer #2: Yes

Response: Thank you.

3. Have the authors made all data underlying the findings in their manuscript fully available?

Reviewer #1: Yes

Reviewer #2: Yes

Response: Thank you.

4. Is the manuscript presented in an intelligible fashion and written in standard English?

Reviewer #1: Yes

Reviewer #2: Yes

Response: Thank you. We proofread our manuscript to check for typographical or grammatical errors.

5. Review Comments to the Author

For Reviewer #1: 

a. First, with regards to the mutations that you engineered into SOD1 to make it more thermostable and exist in the monomeric state, are any of these mutations linked to ALS? In other words, can one consider your engineered SOD1 a 'wild-type' or 'mutant' enzyme with regards to disease? 

Response:

Our engineered SOD1 is considered as a ‘mutant’ enzyme with regards to its altered sequence and slightly reduced activity. The enzyme is properly folded as we demonstrated with our 1D-and 2D-NMR spectra and was active as demonstrated by our in-gel enzymatic assay. As reported by Banci et al., 1997 (reference [13]), the enzyme has the same overall net charge as the wild-type SOD1 but has 8-28-fold reduced activity (depending upon assay) compared to WT. With regard to disease, we could not find an ALS variant with any of these specific mutations (C6A, C111S, F50E, G51E, V148K, and I151K) when we searched the ALSoD website (https://alsod.ac.uk/). Furthermmore, there were no reported SOD1 gene variants for C6, F50, G51 or I151K. There are, however, known mutations at position 111 and 148. These include a C111Y SOD1 gene variant (DOI: 10.1080/17482960801900073; DOI: 10.3109/17482968.2011.656311) and SOD1 gene variants for V148: V148G (DOI: 10.1126/science.8351519) and V148I (DOI: 10.1093/hmg/4.3.491), where the hydrophobic amino acid valine was changed to other hydrophobic amino acids. In the literature, it was reported that mutant SOD1 with C111S (or any of the 4 cycteines changed to serine, [6]), behaved similarly to WT.Overall, because our SOD1 has 6 mutations, is monomeric and has reduced activity, we consider it a mutant protein suitable for NMR experiments. However, it does not carry any mutations that are found in fALS patients. Therefore, with regard to disease, it is not an fALS variant. We inserted this information about our monomeric SOD (being a monomeric mutant enzyme but without fALS mutations) in the introduction of the manuscript (lines 114-121, pages 6-7).

Second, the gel images would be much easier to interpret if the lanes were labeled with their contents, not just numbers.

Response:

Thank you for the suggestion. This is a good idea. We have now inserted text below the numbered lanes so that readers can more easily interpret the results. We also changed the text of the Figure 2 legend, placing ‘– ‘or ’+’ metals for the absence or presence of metals in the culture medium, BI for before induction with IPTG, RT for room temperature, Pe for periplasm, Su for sucrose extract, P for pellets and S for supernatant (pages 21-22).

For Reviewer #2: 

a. This paper should be published without revision. The paper is well-written, well thought out, and its subject is important. The new monomeric form of SOD1 might prove to be useful to certain groups working on SOD1. While I point out a few issues that the authors could address, these are optional.

The paper creates a new monomeric model of SOD1 that will allow easier study with NMR, in particular, the study of protein-ligand interactions. This monomeric form is more stable (offers better yields in recombinant expression systems). These types of SOD1 models, or “Franken-SOD1’s”, as I like to call them, are oligo-mutated forms of SOD1. This one has 6 mutations to make it stable and monomeric, unlike the previous Franken-SOD1’s that had two or four mutations). This one is also not acetylated at the N-terminus (due to the expression in e. coli), so we could add a “seventh” chemical perturbation that moves it further and further away from naturally occurring SOD1 (in addition to the eighth perturbation, an ALS mutation). Here, I must be frank and honest with the authors. When one considers that the subtle switch of an R-OH group on the surface of SOD1, to an R-NH2 group (that is, the D101N mutation), results in one of the most severe forms of ALS, it should remind us to perturb the SOD1 protein as little as possible when studying it. This paper throws that prudent guideline of science completely out the window and speeds away! Their boldness is impressive, but I hope these are the last mutations we'll make for optimization. If you ever need even better yields, I would just scale up, use more flasks, and maybe buy another shaker, instead of adding more mutations. But maybe I am too conservative. I will also say that someone needs to contact David Borchelt at UF and try to convince him to create a transgenic mouse model for some of these Franken-SOD1’s. We need to make sure that they are actually “WT” and do not trigger ALS. For years, we have been working on these proteins, calling them “WT”, “pseudo-WT”. We should probably refer to them as NPMWH variants (Non-Pathogenic Mutants, We Hope)”.

Response: We agree that it would be best to study interactions of ligands with ‘WT’ SOD1 proteins rather than these “Franken-SOD1’s”. We started these investigations using a ‘WT’ SOD1 that contained a His6X-affinity tag fused to the amino-terminus of SOD1 (Grad et al [16]). However, this ‘WT’ protein was poorly expressed in rich or minimal medium with or without added metals and we needed to search for a more suitable protein amenable to our NMR experiments. We selected Banci’s monomer (1997, [13)] as it retained its activity (though several-fold less) and carried the same overall charge as ‘WT’ SOD1. Moreover, because NMR experiments that identify amino acids interactions with ligands require isotopically labelled purified protein, our ‘His6X-WT’ construct was unsuitable for these experiments as we would have needed to grow several litres to purify enough protein. While growing several litres would be doable, producing isotopically labelled protein would be cost-prohibitive as the15NH4Cl needed for 1 litre costs $100 and13C-glucose costs $1000 per litre.

Yes, it would be very informative but expensive and time consuming to create transgenic mice with these peudo-WT SOD1 to hopefully ascertain if they truly are ‘WT’ and non-pathogenic mutants. However, these experiments could take many months to complete and we are required to complete our resubmission by March 5, 2021. 

Issues to consider (optional):

1. Metals! What is the metal content? Is it properly metallated or does it mismetallate? ICP-MS and UV-Vis could help you get the answer quickly.

Response: As the copper ion is necessary for enzymatic activity and the zinc ion helps with the copper coordination and SOD1 structure, and we demonstrated that our monomer was active by an in-gel enzymatic assay, we would conclude that our monomer is properly metallated with copper (for activity) and zinc (for SOD1 structure). We added this detail to the results (line 542-543, page 26) and discussion (line 800, page 39). However, the in-gel assay only provides indirect evidence that our SOD monomer is properly metallated. While we would like to conduct ICP-MS experiments to measure the copper and zinc content of our monomer, the University of Alberta has COVID-19 restrictions in place. From March 2020 to possibly November 2021, only experiments with COVID-19 can be conducted on campus. Therefore, we would not be permitted to run the ICP-MS experiments at this time. While there are different 1H-15N-HSQCs of SOD1 without metals downloaded to the Biological Magnetic Resonance Database, these experiments were conducted at different pHs (pH 5 or 6) compared to our experiments performed at pH 7.4. We could not directly compare these assignments to our HSQC to determine if there was a shift in the histidine and aspartate residues involved in the metal coordination in SOD1.

2. Intro: The authors state that there are conflicts between the works in references 8 and 9 (Lansbury and Hasnain) on whether certain molecules bind SOD1 or not, and that NMR can be an arbiter. They are right, NMR is the right/best tool to get a clear answer. But is there really a conflict between refs 8 and 9? One is an in silico prediction that does not attempt to model explicit solvent (ignoring entropic contributions), also performed at low (physiological) ionic strength, and the other is a crystallographic/experimental study, performed in molar quantities of salt (ammonium sulfate I presume), which increases the hydrophobic (solvent-entropy) effects and hydrophobic interactions that might not happen in physiological settings. Nobody can predict which, whether, and how molecules bind to proteins, de novo. This is why “drug design” is actually high-throughput screening.

Response: We agree that there is no conflict between references 8 and 9 and thank the reviewer for the additional analyses of these studies. We changed ‘conflicting’ to ‘varying’ in our manuscript (line 101, page 5) to reflect this reviewer’s critique. We appreciate this reviewer’s support of NMR-ligand experimentation.

3. Intro: No references on WT enhancing the toxicity of mutant SOD1 through heterodimerization (mutant enhancing WT is mentioned, but several mouse models suggest it might be the other way around!). The rate of mutant/WT sod1 heterodimerization is on the same timescale of its lifetime in vivo, and free energies have been measured for some mutants and metalation states.

Response: We were unaware of experiments that demonstrated that the WT SOD1 enhanced the toxicity of the mutant SOD1. However, as this was mentioned by this reviewer, we found two related references (DOI.org/10.1073/pnas.1902483116; DOI: 10.1093/hmg/ddu517). One of our publications (Grad et al.[16]) demonstrated that the toxic mutant phenotype of SOD1 could be propagated to adjoining cells containing WT SOD1. Our experience with SOD1 suggested that the toxic phenotype was propagated from mutant to “WT’ in vitro. As this reviewer pointed out, WT SOD1 also can add to the pathogenesis of ALS by forming heterodimers with mutant SOD1, effectively removing both from the SOD1 pool within the cell. Additionally, these heterodimers form inclusions within cells that affect the longevity of these cells, possibly accelerating ALS pathogenesis (DOI.org/10.1073/pnas.1902483116). From a bigenic transgenic mouse study (DOI: 10.1093/hmg/ddu517), it was concluded that this heterodimerization could vary with SOD1 mutations and was not consistent across all mutant SOD1 bigenic transgenic crosses. Therefore, one would conclude that the WT enhancing the toxicity of SOD1 mutants occurs but not consistently with all mutants. However, as mentioned in our introduction: the fact that the mutant SOD1 interacts with ‘WT’SOD, converting it to a toxic phenotype, could be concluded from these heterodimerization experiments too. Because this information was difficult to incorporate into our introduction without changing all the reference numbering in the original document, we have decided not to add this information. However, we do thank this reviewer for expanding our knowledge of the potential mechanisms of ALS pathogenesis though interactions of ‘WT’ SOD1 with mutant SOD1. ________________________________________

6. PLOS authors have the option to publish the peer review history of their article (what does this mean?). If published, this will include your full peer review and any attached files.

Do you want your identity to be public for this peer review? For information about this choice, including consent withdrawal, please see our Privacy Policy.

Reviewer #1: No

Reviewer #2: No

Once again, we would like to thank the reviewers and the editors for their comments and suggestions. We believe these suggested changes have improved the manuscript. I hope that the changes we have made are satisfactory and we are looking forward to seeing the manuscript published in PLoS ONE soon.

Sincerely,

David Wishart

---

## [Editor Report · Decision Letter 1]

11 Feb 2021

Cloning and high-level expression of monomeric human superoxide dismutase 1 (SOD1) and its interaction with pyrimidine analogs

PONE-D-20-39700R1

Dear Dr. Wishart,

We’re pleased to inform you that your manuscript has been judged scientifically suitable for publication and will be formally accepted for publication once it meets all outstanding technical requirements.

Kind regards,

Oscar Millet

Academic Editor

PLOS ONE
---

## [Editor Report · Acceptance letter]

15 Feb 2021

PONE-D-20-39700R1 

Cloning and high-level expression of monomeric human superoxide dismutase 1 (SOD1) and its interaction with pyrimidine analogs 

Dear Dr. Wishart:

I'm pleased to inform you that your manuscript has been deemed suitable for publication in PLOS ONE. Congratulations! Your manuscript is now with our production department. 

Kind regards, 

on behalf of

Dr. Oscar Millet 

Academic Editor

PLOS ONE